# Neuronal SphK1 acetylates COX2 and contributes to pathogenesis in a model of Alzheimer's Disease

Ju Youn Lee[1,2,3], Seung Hoon Han[1,2,3], Min Hee Park[1,2,3], Bosung Baek[1,4], Im-Sook Song[5], Min-Koo Choi[6], Yoh Takuwa[7], Hoon Ryu[8,9], Seung Hyun Kim[10], Xingxuan He[11], Edward H. Schuchman[11], Jae-Sung Bae[1,2,3] & Hee Kyung Jin[1,4]

Although many reports have revealed the importance of defective microglia-mediated amyloid β phagocytosis in Alzheimer's disease (AD), the underlying mechanism remains to be explored. Here we demonstrate that neurons in the brains of patients with AD and AD mice show reduction of sphingosine kinase1 (SphK1), leading to defective microglial phagocytosis and dysfunction of inflammation resolution due to decreased secretion of specialized proresolving mediators (SPMs). Elevation of SphK1 increased SPMs secretion, especially 15-R-Lipoxin A4, by promoting acetylation of serine residue 565 (S565) of cyclooxygenase2 (COX2) using acetyl-CoA, resulting in improvement of AD-like pathology in APP/PS1 mice. In contrast, conditional SphK1 deficiency in neurons reduced SPMs secretion and abnormal phagocytosis similar to AD. Together, these results uncover a novel mechanism of SphK1 pathogenesis in AD, in which impaired SPMs secretion leads to defective microglial phagocytosis, and suggests that SphK1 in neurons has acetyl-CoA-dependent cytoplasmic acetyl-transferase activity towards COX2.

[1] Stem Cell Neuroplasticity Research Group, Kyungpook National University, Daegu 41566, South Korea. [2] Department of Physiology, Cell and Matrix Research Institute, School of Medicine, Kyungpook National University, Daegu 41944, South Korea. [3] Department of Biomedical Science, BK21 Plus KNU Biomedical Convergence Program, Kyungpook National University, Daegu 41944, South Korea. [4] Department of Laboratory Animal Medicine, College of Veterinary Medicine, Kyungpook National University, Daegu 41566, South Korea. [5] College of Pharmacy and Research Institute of Pharmaceutical Sciences, Kyungpook National University, Daegu 41566, South Korea. [6] College of Pharmacy, Dankook University, Cheon-an 31116, South Korea. [7] Department of Physiology, Kanazawa University School of Medicine, Kanazawa, Ishikawa 920-8640, Japan. [8] VA Boston Healthcare System, Department of Neurology and Boston University Alzheimer's Disease Centre, Boston University School of Medicine, Boston, MA 02130, USA. [9] Centre for Neuromedicine, Brain Science Institute, Korea Institute of Science and Technology, Seoul 02792, South Korea. [10] Department of Neurology, Hanyang University College of Medicine, Seoul 04763, South Korea. [11] Department of Genetics and Genomic Sciences, Icahn School of Medicine at Mount Sinai, New York, NY 10029, USA. Correspondence and requests for materials should be addressed to J.-S.B. (email: jsbae@knu.ac.kr) or to H.K.J. (email: hkjin@knu.ac.kr)

Alzheimer's disease (AD), the most common form of dementia, is characterized by accumulation of extracellular amyloid plaques and intracellular neurofibrillary tangles composed of aggregated amyloid β (Aβ) and hyperphosphorylated tau, respectively, leading to progressive cognitive impairment and dementia[1]. Besides these hallmarks, dysregulation of glial cells, especially microglia that normally associate closely with Aβ, also are observed[2]. In the AD brain, dysfunctional microglia contribute to the disease progression by releasing pro-inflammatory cytokines in response to Aβ deposition and/or by decreased Aβ phagocytosis[3–6]. Dysfunction of microglia also causes chronic inflammation that is a consequence of failure to resolve the inflammation. Resolution is the final stage of the inflammatory response, and is induced by specialized proresolving mediators (SPMs) including Lipoxin A4 (LxA4), Resolvin E1 (RvE1), and Resolvin D1 (RvD1), known to be produced in microglia[7,8]. This results in restoration of microglia function, such as increase of phagocytic microglia, downregulation of pro-inflammatory cytokines, and active clearance of apoptotic cells and debris[1,7]. Importantly, a recent study has reported that the resolution of inflammation is dysfunctional in patients with AD[9].

Sphingosine kinase (SphK) types 1 and 2 are key enzymes that convert sphingosine into sphingosine-1-phosphate (S1P), bioactive lipids known to regulate inflammation. Recently, the role of SphK in inflammation has been the main focus of research on many diseases, such as asthma, inflammatory bowel disease, and rheumatoid arthritis[10]. However, the roles of SphK in neuroinflammation have not been fully understood in the AD brain. Here we show that neuronal SphK1 levels are reduced in AD brain, leading us to investigate the role of neuron-derived SphK1 in AD pathogenesis. Increased neuronal SphK1 improved microglia-mediated Aβ phagocytosis and protected cognitive deficits. Neuronal SphK1 also promoted SPMs secretion, especially 15-R-LxA4, by acetylating serine residue 565 (S565) of cyclooxygenase2 (COX2), resulting in increase of phagocytic microglia. Furthermore, we demonstrate that SphK1 activation restores SPMs secretion in AD patient-specific neurons, suggesting an important role of neuronal SphK1 as a regulator of microglial phagocytosis in AD, and indicating its potential therapeutic use for AD.

## Results

**Neuronal SphK1 affects AD pathology in APP/PS1 mice.** To examine the role of SphK in AD, we first checked the SphK levels in cortex of APP/PS1 mouse brains. *SphK1* mRNA expression and SphK activity were decreased in the brain of APP/PS1 mice compared with those of wild-type (WT) mice, while *SphK2* mRNA did not differ between the groups (Fig. 1a). No significant differences in sphingosine and S1P levels were found (Fig. 1b). Next, to further investigate the cell contribution of decreased SphK1 levels in mouse brains, we isolated neurons, microglia, and astrocytes, known to be important cells types related to AD pathogenesis (Supplementary Fig. 1). *SphK1* mRNA expression and SphK activity were significantly decreased in APP/PS1 neurons compared with WT neurons, although there were no significant difference in microglia and astrocytes. *SphK2* mRNA expression did not differ in neurons, microglia, and astrocytes derived from WT and APP/PS1 mice (Fig. 1c). Sphingosine and S1P levels also were unchanged in these cells (Fig. 1d).

To further validate the reason of unaltered lipid levels in spite of reduced SphK activity in APP/PS1 mice, we investigated the relationship between SphK activity and lipid levels using neurons derived from WT, $SphK1^{-/-}$, $SphK2^{-/-}$, and SphK1 tg mice. SphK activity was decreased in $SphK1^{-/-}$ and $SphK2^{-/-}$

mice-derived neurons and increased in SphK1 tg mice-derived neurons compared with neurons derived from WT mice. Particularly, SphK activity in $SphK1^{-/-}$ mice-derived neurons was lower than $SphK2^{-/-}$, indicating SphK1 was more effective in SphK activity. Unlike SphK activity, we found that S1P was dramatically decreased in neurons derived from $SphK2^{-/-}$ mice, but not in neurons derived from $SphK1^{-/-}$ and SphK1 tg mice. Sphingosine levels were not different between the groups (Fig. 1e), similar to a previous study[11]. These results suggested that SphK activity was principally dependent on SphK1 expression, and lipid levels, such as S1P, were principally regulated by SphK2 expression. Therefore, the comparable lipid levels in APP/PS1 mice and WT might be due to comparable SphK2 expression. Consistent with the presented data, SphK1 level in immunoreactivity and western blotting also was markedly reduced in APP/PS1 neurons (Fig. 1f, g). These results confirmed that SphK1 was diminished in APP/PS1 neurons and that this reduction, regardless of the lipid alternations, may influence disease progression and/or pathogenesis.

Next, to assess the influence of SphK1 in AD pathology, we bred SphK1 overexpressing tg mice to APP/PS1 animals (Supplementary Fig. 2a, b). In cortex derived from APP/PS1/SphK1 tg mice brain, SphK1 levels were increased (Fig. 1a). Importantly, *SphK1* mRNA expression and SphK activity also were significantly increased in the neurons of APP/PS1/SphK1 tg mice compared with those of APP/PS1 mice. *SphK1* mRNA levels were increased in the microglia and astrocytes of APP/PS1/SphK1 tg mice as well, but SphK activity was not altered (Fig. 1c). Similar to previous results, other sphingolipid factors were unaltered in the APP/PS1/SphK1 tg mice in spite of SphK1 overexpression (Fig. 1b, d). Immunofluorescence and western blotting also indicated restoration of SphK1 expression in neurons of APP/PS1/SphK1 tg mice (Fig. 1f, g).

To determine whether the increased neuronal SphK1 activity in APP/PS1/SphK1 tg mice affected AD pathology, we first determined the Aβ profile. Thioflavin S (ThioS) staining, immunofluorescence of 6E10, Aβ40 and Aβ42, and ELISA of Aβ40 and Aβ42 showed significantly lower Aβ levels in the 9-month-old APP/PS1/SphK1 tg mice than age-matched APP/PS1 mice (Fig. 2a, b and Supplementary Fig. 2c–e). In APP/PS1/SphK1 tg mice, cerebral amyloid angiopathy also was reduced (Supplementary Fig. 2f). There were no significant differences of tau hyperphosphorylation between the two groups (Supplementary Fig. 2g). Synaptophysin, MAP2, Synapsin1, and PSD95 labeling density were reduced in APP/PS1 mice versus WT mice. However, APP/PS1/SphK1 tg mice had labeling density that was restored to those of WT mice (Supplementary Fig. 2h–k).

We also performed the Morris water maze task and fear conditioning to examine changes of learning and memory. Aged APP/PS1 mice showed severe deficits in memory formation, while APP/PS1/SphK1 tg mice were largely protected from this defect (Supplementary Fig. 3a–h). For locomotion and spontaneous activity, an open field test was used. APP/PS1/SphK1 tg mice showed improved locomotion and spontaneous activity compared with APP/PS1 mice (Supplementary Fig. 3i–l). Collectively, these results suggested that increased neuronal SphK1 in APP/PS1 mice reduced the Aβ load and improved learning and memory.

**Neuronal SphK1 modulates neuroinflammation in APP/PS1 mice.** To confirm the mechanism responsible for restoration of AD pathology in APP/PS1/SphK1 tg mice, we first assessed the apoptotic responses and APP processing in brain, but did not detect differences between the two groups (Supplementary Fig. 4a, b). Next, we examined the Aβ generating enzyme Bace-1,

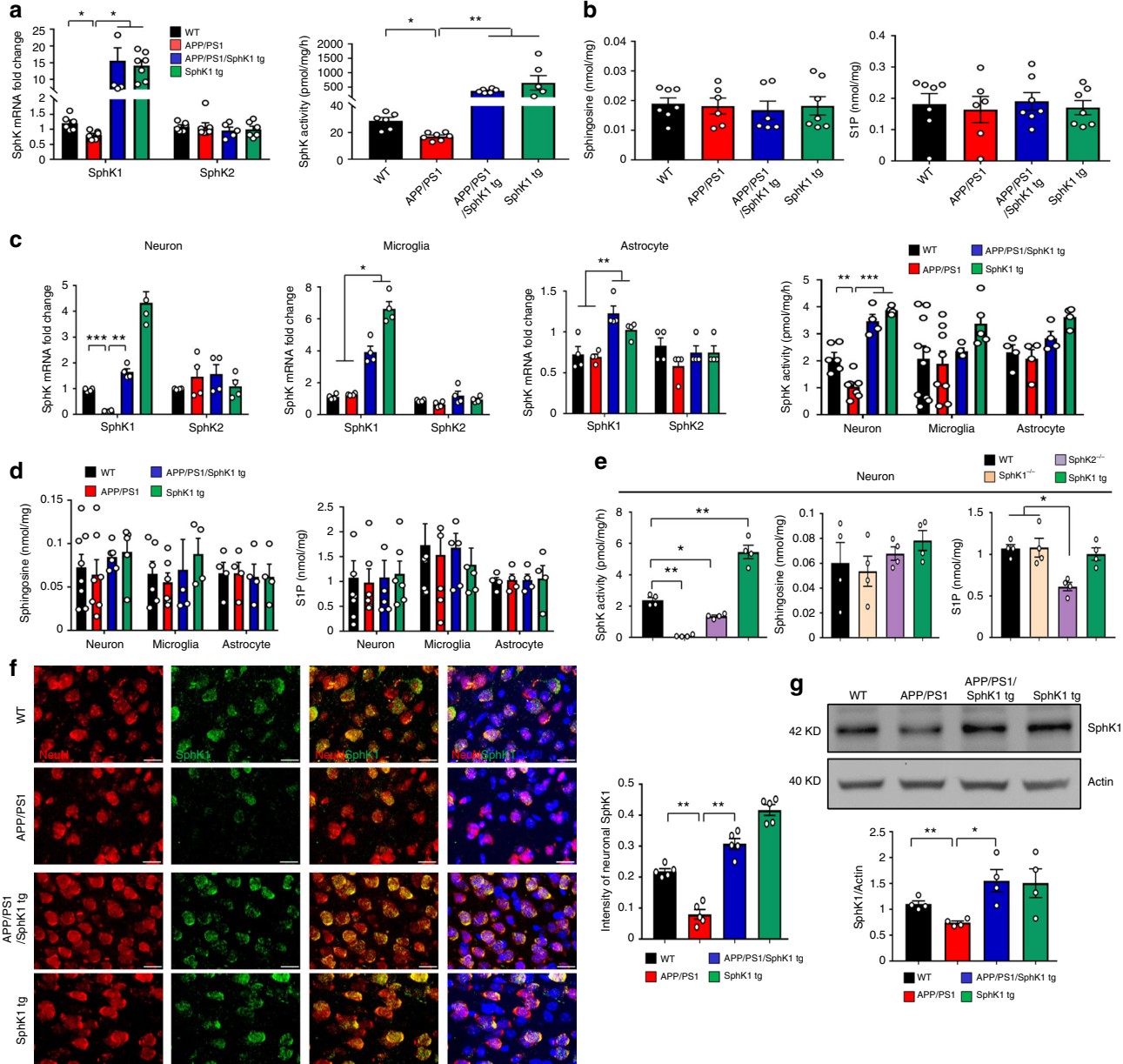

**Fig. 1** SphK1 is decreased in APP/PS1 mice neuron. **a** *SphK1* ($n = 6$–7 per group) and *SphK2* mRNA ($n = 5$–7 per group) and SphK activity ($n = 6$–7 per group) in cortex of brain. **b** Detection of sphingosine ($n = 6$–7 per group) and S1P ($n = 6$–7 per group) in brain. **c** *SphK1* and *SphK2* mRNA ($n = 4$ per group) and SphK activity ($n = 4$–10 per group) in neurons, microglia, and astrocytes isolated from mouse brain. **d** Detection of sphingosine ($n = 4$–8 per group) and S1P ($n = 4$–8 per group) in neurons, microglia, and astrocytes isolated from mouse brain. **e** Detection of SphK activity, sphingosine and S1P in neurons isolated from WT, *SphK1*$^{-/-}$, *SphK2*$^{-/-}$ and SphK1 tg mouse brain ($n = 4$ per group). **f** Left, representative immunofluorescence images of cortex showing SphK1 (green) merged with neuron (NeuN, red). Scale bars, 20 μm. Right, quantification of neuronal SphK1 ($n = 5$ per group). **g** Western blotting for SphK1 in neuron isolated from mouse brain ($n = 4$ per group). All data analysis was performed on 9-month-old mice. **a**–**g** One-way analysis of variance, Tukey's post hoc test. *$P < 0.05$, **$P < 0.01$, ***$P < 0.001$. All error bars indicate s.e.m.

but did not detect differences (Supplementary Fig. 4c). Also, the expression pattern of proteins related to endocytosis and autophagy showed no differences between the groups (Supplementary Fig. 4d, e). Our results indicated that apoptosis, processing of APP, modulation of the endocytic pathway and autophagy were not related with the increase of SphK1 in the APP/PS1/SphK1 tg mouse.

Inflammatory processes and the innate immune response are activated in the brains of people with AD[1]. In principle, this neuroinflammation could either drive pathology or be the result of the ongoing disease process. To investigate the effect of increased neuronal SphK1 on neuroinflammation, we examined changes of microglia and astrocytes. APP/PS1/SphK1 tg mice showed a significant reduction of activated microglia and astrocytes compared to APP/PS1 mice (Fig. 2c, d). In addition, APP/PS1/SphK1 tg mice showed a decrease of pro-inflammatory markers, including *TNF-α, IL-1β, IL-6* and *iNOS*, and immunoregulatory cytokine *IL-10*, and an increase of anti-inflammatory markers, including *IL-4, TGF-β* and *Arg1*, compared with APP/PS1 mice (Fig. 2e). Collectively, these data indicated that genetic *SphK1* overexpression could improve the inflammatory response in AD brains, and that regulation of neuroinflammation might be an important mechanism that attenuates AD pathology by increased neuronal SphK1.

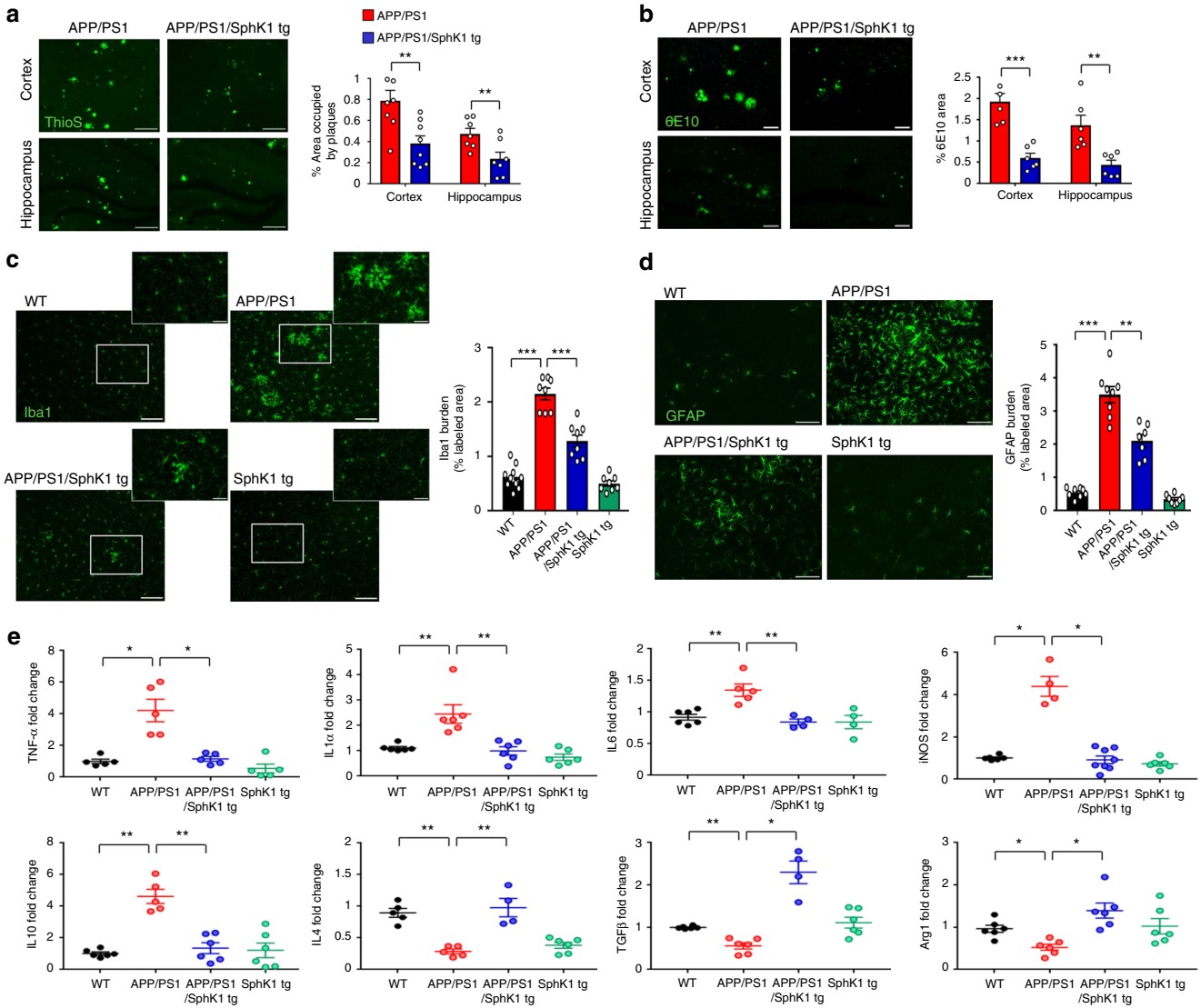

**Fig. 2** Elevation of SphK1 reduces Aβ deposition and neuroinflammation in APP/PS1 mice. **a** Left, representative immunofluorescence images of thioflavin S (ThioS, Aβ plaques) in cortex and hippocampus of APP/PS1 and APP/PS1/SphK1 tg mice. Scale bars, 200 μm. Right, quantification of area occupied by Aβ plaques ($n = 7$–8 per group). **b** Representative immunofluorescence images and quantification of 6E10 ($n = 5$–6 per group; Scale bars, 80 μm). **c** Left, immunofluorescence images of microglia (Iba1) in cortex of WT, APP/PS1, APP/PS1/SphK1 tg or SphK1 tg mice brain. Low-magnification scale bars, 100 μm; High-magnification scale bars, 30 μm. Right, quantification of microglia ($n = 8$–10 per group). **d** Left, immunofluorescence images of astrocyte (GFAP) in cortex of mice brain. Scale bars, 100 μm. Right, quantification of astrocytes ($n = 7$–8 per group). **e** mRNA levels of inflammatory markers in cortex of mice brain ($n = 4$–8 per group). Pro-inflammatory marker: *TNF-α, IL-1β, IL-6 and iNOS*, Immunoregulatory cytokine: *IL-10*, Anti-inflammatory marker: *IL-4, TGF-β and Arg1*. All data analysis was performed on 9-month-old mice. **a**, **b** Student's *t* test. **c**–**e** One-way analysis of variance, Tukey's post hoc test. *$P < 0.05$, **$P < 0.01$, ***$P < 0.001$. All error bars indicate s.e.m.

**Reduced neuronal SphK1 by Aβ activates microglia**. Next, to investigate the relationship of Aβ and SphK1, we examined the time course of amyloid deposition in relation to SphK1 levels in APP/PS1 mice. ThioS staining showed that Aβ plaques began to appear at the age of 6 months (Supplementary Fig. 5a). Neuronal SphK1 also started to decrease in six-month-old APP/PS1 mice, and was significantly reduced at 9 months (Supplementary Fig. 5b). Microglia and astrocytes were activated at the age of 9 months in APP/PS1 mice (Supplementary Fig. 5c, d). Collectively, these results indicated that reduction of neuronal SphK1 was correlated with Aβ deposition, leading to activated glia cells in APP/PS1 mice.

To interrogate more directly the relationship of Aβ and SphK1, we treated neurons with 10 μM Aβ and determined the SphK1 levels. Treatment with Aβ reduced *SphK1* mRNA expression and SphK activity in neurons (Supplementary Fig. 5e). *SphK1* mRNA

expression is regulated through PI3K, AKT and mTOR[12] signaling, and the enzyme activity of SphK1 is modulated by ERK1/2-mediated phosphorylation[13]. To determine the signaling molecules that modulate SphK1 expression and activity in Aβ-treated neurons, we examined the levels of PI3K and phospho-ERK. PI3K levels showed a dramatic decrease following addition of Aβ to neurons, although no significant difference was observed in p-AKT levels (Supplementary Fig. 5f). Phospho-ERK levels also were not different between the two groups (Supplementary Fig. 5g). Moreover, a specific pharmacological activator of the PI3K signaling cascade (740YP)[14] increased SphK1 levels in Aβ-treated neurons, while an inhibitor of PI3K signaling (rapamycin)[12] reduced it (Supplementary Fig. 5h). These data showed that Aβ led to reduction of neuronal SphK1 through the PI3K signaling pathway.

Next, to define the glial cell types responsible for neuronal SphK1-mediated inflammation, we utilized conditioned media (CM) from isolated primary neurons, microglia, and astrocytes[15]. Neurons were stimulated with Aβ and CM was harvested as described in Supplementary Fig. 6a, b. CM of the Aβ-treated neurons induced a significant shift of inflammatory cytokines in microglia, while it had much less effect on the astrocytes (Supplementary Fig. 6a, b). Intriguingly, further transfer of the CM from these microglia to astrocytes led to significant changes in the production of inflammatory factors (Supplementary Fig. 6a). However, in the converse situation, when exposed to Aβ-treated neuron-astrocyte-CM, the microglia did not become activated (Supplementary Fig. 6b). Thus, the sequence of microglial activation appears to be neurons to microglia directly, not through astrocytes.

To elucidate whether neuronal SphK1 directly controlled microglial activation, neurons derived from WT and SphK1 tg mice were stimulated with Aβ, and CM was harvested as described in Supplementary Fig. 6c. Microglial activation was reduced after treatment of CM from Aβ-stimulated SphK1 tg neuron compared with CM from Aβ-treated WT neuron. Similar results were observed in astrocytes induced with Aβ-treated neuron-microglia-CM (Supplementary Fig. 6c). However, astrocytes were not directly affected by either Aβ-treated WT or SphK1 tg neurons CMs (Supplementary Fig. 6d). We also confirmed the visible differences in the response of microglia to Aβ-treated WT and SphK1 tg neurons. Morphology of activated microglia was observed in microglia treated with CM from Aβ-treated WT neurons compared with control neurons, and was reduced in microglia treated with CM from Aβ-treated SphK1 tg neurons (Supplementary Fig. 6e). These data indicated that microglia was more responsive than astrocytes to neuronal SphK1-mediated reduction of inflammation.

**Neuronal SphK1 regulates SPMs secretion by acetylating COX2.** Recent findings indicate that neurons are not merely passive partners of microglia, but rather control microglial activity. The variety of different signals that neurons use to control microglia can be divided into two categories: "Off" signals constitutively keep microglia in their resting state, antagonize pro-inflammatory activity, and include CD47, CX3CL1, BDNF, and CD200, and "On" signals are inducible and include CXCL10, CCL21, and MMP3[16]. To test whether neurons with increased SphK1 regulate microglia activity through Off/On signals, we assessed these molecules in neuron derived from these mice. Although APP/PS1 neurons showed reduction of Off signals and increase of On signals compared with WT neurons, this did not change in APP/PS1/SphK1 tg neurons (Supplementary Fig. 7).

SPMs are a recently identified group of molecules that function in the resolution of inflammation. This includes arachidonic acid (AA)-derived LxA4, eicosapentaenoic acid (EPA)-derived RvE1 and docosahexaenoic acid (DHA)-derived RvD1. SPMs are agonists that can stimulate key cellular resolution events such as limiting neutrophil infiltration and enhancing clearance of debris by macrophages and microglia[7,17,18]. We therefore examined whether neuronal SphK1 induces the secretion of SPMs regulating microglial activation, using ELISA. LxA4 and RvE1 levels were significantly lower in CM derived from APP/PS1 neurons than WT neurons and recovered in CM derived from APP/PS1/SphK1 tg neurons, suggesting that neuronal SphK1 might be related with secretion of SPMs (Fig. 3a). However, RvD1 did not change.

SPMs are biosynthesized from essential polyunsaturated fatty acids, such as AA, EPA, and DHA, via the consecutive actions of lipoxygenase (LOX-15) and/or COX2 enzymes[7,19]. We therefore tested levels of LOX-15 and COX2 in neurons derived from mouse brains. The mRNA and protein levels of these enzymes did not vary between the groups (Fig. 3b, c). Recent studies have also revealed that aspirin inhibits prostanoid biosynthesis by acetylation of COX1 and COX2. Acetylation of COX2 by aspirin modifies the catalytic domain, blocking prostaglandin biosynthesis, but it remains active and produces SPMs in cells carrying COX2[20]. Based on this concept, to examine whether SphK1 promotes acetylation of COX2 using acetyl-CoA in neurons, we purified COX2 and measured the acetylation level in the presence of [14C] acetyl-CoA using neuronal lysates derived from 9-month-old mice. Compared with WT, COX2 displayed low-level acetylation in neuron of APP/PS1 mice and this was enhanced in neuron of APP/PS1/SphK1 tg mice, suggesting that SphK1 modulates COX2 acetylation in neurons using acetyl-CoA (Fig. 3d).

Next, for confirmation of COX2 acetylation-derived SPMs secretion by SphK1, we identified several SPMs, including 15-R-LxA4, 15-S-LxA4, 17-R-RvD1, 17-S-RvD1, 18-R-RvE1 and PD1, using systematic LC-MS/MS. Interestingly, only 15-R-LxA4, one of COX2 acetylation-derived products[21], were found in neurons. Neurons derived from APP/PS1 mice exhibited a marked decrease of 15-R-LxA4 compared with neurons derived from WT mice. These were restored in neurons derived from APP/PS1/SphK1 tg mice (Fig. 3e, f). 15-S-LxA4, 17-R-RvD1, 17-S-RvD1, 18-R-RvE1, and PD1 were not detected in neurons. Of note, because LC-MS/MS was based on the exact molecular formula rather than a fragmentation pattern, SPMs levels measured by LC-MS/MS was significantly lower or not detected compared with levels measured by ELISA. Overall, we confirmed that 15-R-LxA4, which is reported to be decreased in AD[9], was reduced in APP/PS1 mice, and that the elevation of SphK1 increased COX2 acetylation-derived 15-R-LxA4 secretion, suggesting the potential therapeutic use of SphK1 in AD. To gain more direct insights into the relationship between SphK1 and SPMs secretion in neurons, we treated WT neurons with *SphK1* siRNA and determined the changes of SPMs. CM derived from *SphK1* siRNA-treated neurons showed strongly reduced SPMs levels (except RvD1), similar to CM from APP/PS1 neurons (Supplementary Fig. 8a, b). Knockdown of SphK1 in normal neurons also caused reduction of COX2 acetylation and 15-R-LxA4 (Supplementary Fig. 8c, d).

These findings on SphK1-mediated COX2 acetylation led us to search for possible therapeutic applications of this pathway in APP/PS1 mice. To increase SphK1 in neurons of APP/PS1 mice, we undertook PI3K signal activation using 740YP described above. 740YP-treated neurons derived from APP/PS1 mice exhibited an increase of COX2 acetylation compared with vehicle-treated animals (Supplementary Fig. 9a). Secreted SPMs also were elevated after 740YP treatment (Supplementary Fig. 9b). Together, these data indicated that SphK1 promoted acetyl-CoA-dependent acetyltransferase activity on COX2 in neurons and induced the secretion of SPMs, especially 15-R-LxA4.

**SphK1 is an acetyl-CoA dependent acetyltransferase of COX2.** To investigate the acetyltransferase activity of SphK1, kinetic and functional analysis, including determination of the acetyl binding and dissociation from the enzyme, was performed. Incorporation of acetyl group to SphK1 became saturated with increased concentration of acetyl-CoA, yielding $K_M$ and $K_{cat}$ values of 58.2 μM and 0.0185 min$^{-1}$, respectively (Fig. 4a). Following equilibrium dialysis experiments, the incorporated acetyl group also was dissociated from the acetyl-CoA:SphK1 complex in the presence of competitive free acetyl-CoA in a concentration-dependent manner. This dissociation of acetyl-CoA and SphK1 was also saturated with higher inhibitor concentrations, resulting in a $K_D$

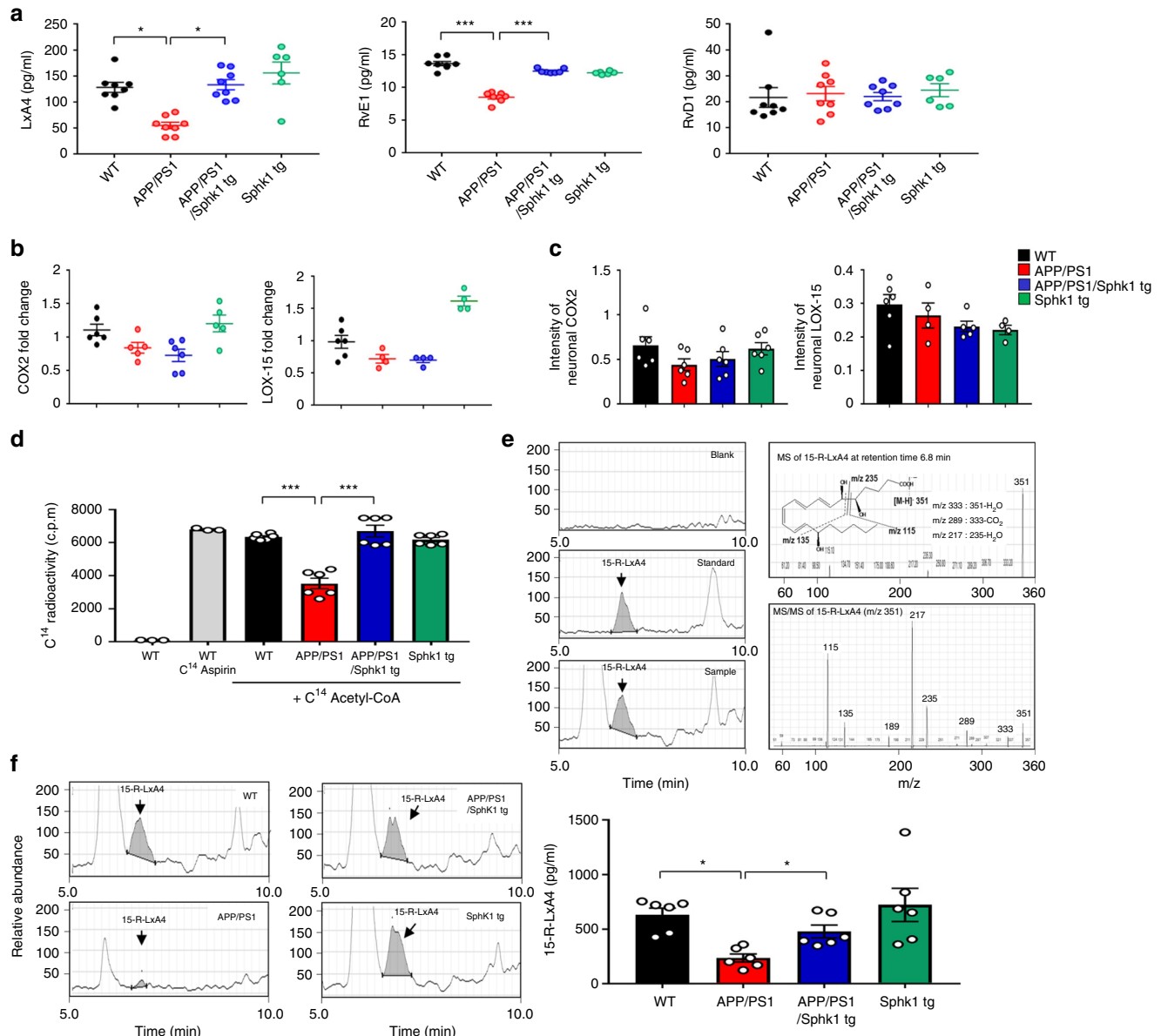

**Fig. 3** Neuronal SphK1 induces SPMs secretion by COX2 acetylation. **a** Protein levels of LxA4, RvE1, and RvD1 were detected by using ELISA in CM of neurons derived from WT, APP/PS1, APP/PS1/SphK1 tg, and SphK1 tg mice ($n = 6$–$8$ per group). **b** mRNA levels of *COX2* and *LOX-15* in neurons derived from cortex of WT, APP/PS1, APP/PS1/SphK1 tg, and SphK1 tg mice ($n = 4$–$6$ per group). **c** Quantification of neuronal COX2 ($n = 6$ per group) and neuronal LOX-15 ($n = 4$–$6$ per group). **d** Acetylation assay of COX2 protein in neurons derived from WT, APP/PS1, APP/PS1/SphK1 tg, and SphK1 tg mice. [$^{14}$C] aspirin-treated neuron was positive control. Sonicated neurons incubated in the presence of [$^{14}$C] acetyl-CoA for 2 h at 37 °C and then COX2 was purified and analyzed on scintillation counter ($n = 3$–$6$ per group). **e** Representative chromatograms of blank, 15-R-LxA4 standard, and 15-R-LxA4 in WT samples (left panel). Molecular MS scanning from the peak at retention time 6.8 min (right upper panel) and MS/MS fragmentation pattern of 15-R-LxA4 from the peak at retention time 6.8 min (right lower panel). **f** Representative chromatograms (left panel) and quantification of 15-R-LxA4 in neurons derived from WT, APP/PS1, APP/PS1/SphK1 tg, and SphK1 tg mice with acetyl-coA treatment (24 h after 2.5 mM acetyl-CoA treatment) (right panel, $n = 6$ per group). All data analysis was done on 9-month-old mice. **a**–**d**, **f** One-way analysis of variance, Tukey's post hoc test. *$P < 0.05$, ***$P < 0.001$. All error bars indicate s.e.m.

value of 6.8 μM (Fig. 4b). The lower $K_D$ (i.e., dissociation constant) value compared with $K_M$ value (i.e., binding affinity) suggested the acetyltransferase characteristics of SphK1. The calculated turnover rate for SphK1 ($K_{cat} = 0.0185$ min$^{-1}$) is quite low in comparison to those of known nuclear acetyltransferases. Since SphK1 is a cytoplasmic enzyme, it is comparable to that recently reported for cytoplasmic tau-K18 ($K_{cat} = 0.005$ min$^{-1}$) and α-tubulin acetyltransferase (αTAT-1, $K_{cat} = 0.037$ min$^{-1}$)[22], which indicates distinct kinetic differences between nuclear and cytoplasmic acetyltransferases. We also confirmed that neuronal

SphK1 acetyltransferase activity depends on concentration of sphingosine, as known substrate of SphK1. Incorporation of the acetyl group into SphK1 was not observed for 0 μM of sphingosine, and increased with increasing concentration of sphingosine, yielding the 60.2 and 6.4 μM $K_M$ values at 10 and 100 μM sphingosine, respectively. However, $K_{cat}$ did not differ with changing concentration of sphingosine (Fig. 4c). These results indicated that both sphingosine and acetyl-CoA were needed in SphK1 acetyltransferase activity, suggesting that sphingosine may participate in this catalytic reaction.

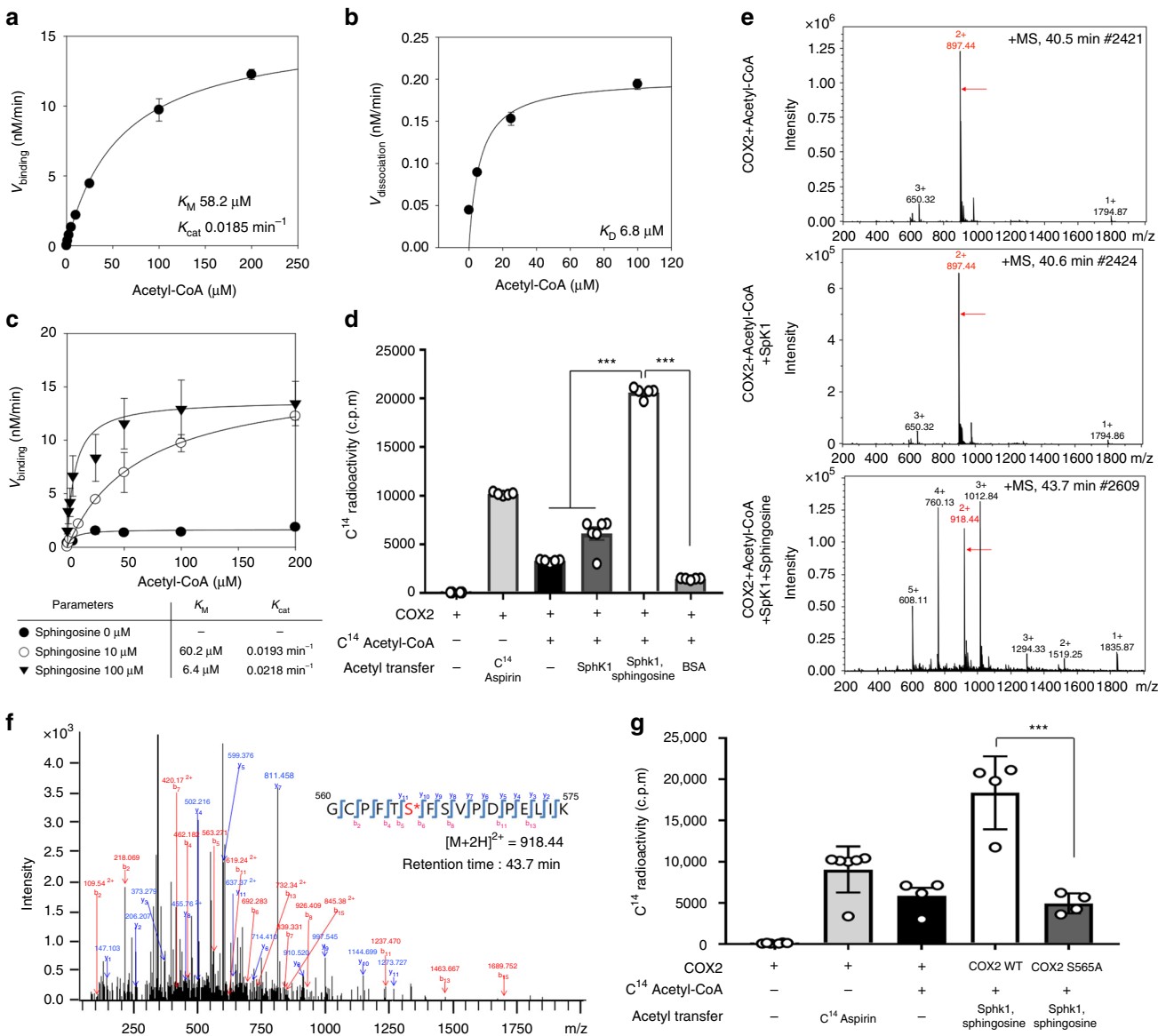

**Fig. 4** SphK1 plays a role of acetyltransferase, acetylating S565 of COX2 in neurons. **a** Acetyl-CoA binding activity of SphK1 was analyzed by filter binding assay with 10 μM sphingosine. The binding velocity ($V_{binding}$) of [³H] acetyl-CoA to SphK1 was plotted to the acetyl-CoA concentration and the nonlinear regression analysis of the saturated plot yielded the kinetic parameters such as $K_{cat}$ (catalytic constant) and $K_M$ (Michaelis−Menten constant) for acetyl-CoA and SphK1 binding activity ($n = 3$ per group). **b** Dissociation of acetyl group from SphK1 was analyzed by equilibrium dialysis in the presence of 0, 5, 25, and 100 μM free acetyl-CoA. The dissociation rate ($V_{dissociation}$) of [³H] acetyl group from [³H] acetyl-CoA and SphK1 complex was plotted against inhibitor-free acetyl-CoA concentration and dissociation constant ($K_D$) was calculated from the nonlinear regression analysis ($n = 3$ per group). **c** Acetyl-CoA binding activity of SphK1 was analyzed by filter binding assay in the presence of 0, 10, and 100 μM sphingosine ($n = 3$ per group). **d** Acetylation assay of purified COX2 protein treated with SphK1 and [¹⁴C] acetyl-CoA in the presence of 100 μM sphingosine or not. BSA-treated COX2 protein was negative control ($n = 4-6$ per group). **e** LC-MS spectra of peptide 560-GCPFTSFSVPDPELIK-575 ($m/z = 918.44$) of COX2 acetylated by SphK1, acetyl-CoA and sphingosine. **f** LC-MS/MS spectra of ac-S565 in 560-GCPFTSFSVPDPELIK-575 of COX2. **g** Acetylation assay of COX2 WT and COX2 S565A recombinants treated with SphK1 and [¹⁴C] acetyl-CoA in the presence of 100 μM sphingosine ($n = 4-6$ per group). **d** One-way analysis of variance, Tukey's post hoc test. **g** Student's $t$ test. ***$P < 0.001$. All error bars indicate s.e.m.

Next, to further characterize this reaction with regard to COX2, purified SphK1 was incubated with COX2 and [¹⁴C] acetyl-CoA in the presence or absence of sphingosine. These results showed that SphK1 possessed higher acetyltransferase activity towards COX2 in the presence of sphingosine rather compared with SphK1 alone, indicating that SphK1 might induce the acetylation on COX2 through sphingosine or a sphingosine intermediate (Fig. 4d).

Finally, to identify site(s) on COX2 acetylated by SphK1, we treated SphK1, acetyl-CoA and sphingosine with COX2. As

mentioned above, COX2 treated with SphK1, acetyl-CoA, and sphingosine had an acetyl group, while COX2 treated in the absence of sphingosine did not. Serine 565 (S565) on peptide 560-GCPFTSFSVPDPELIK-575 of COX2 was identified to be acetylated in the presence of SphK1, and this was different from the acetylated residue by aspirin (S516)[23] (Fig. 4e, f). To establish the causality of this relationship, we mutated S565 of COX2 to an Ala 565 residue (S565A) and performed the acetylation assay. Although COX2 was acetylated by SphK1 and sphingosine, the S565A mutant decreased significantly acetylation

of COX2 in the presence of SphK1, indicating that S565 is the main target site of SphK1-mediated COX2 acetylation (Fig. 4g).

**SPMs from neuronal SphK1 control microglial Aβ phagocytosis.** Previous study reported that SPMs regulate macrophage and rebalance inflammation to promote Aβ phagocytosis[24]. First, to determine whether increased neuronal SphK1 restores microglial recruitment to Aβ, we quantified the number of microglia surrounding the plaques, and found that microglial recruitment was increased in APP/PS1/SphK1 tg mice compared to APP/PS1 mice (Fig. 5a). We next performed a phagocytosis assay using live

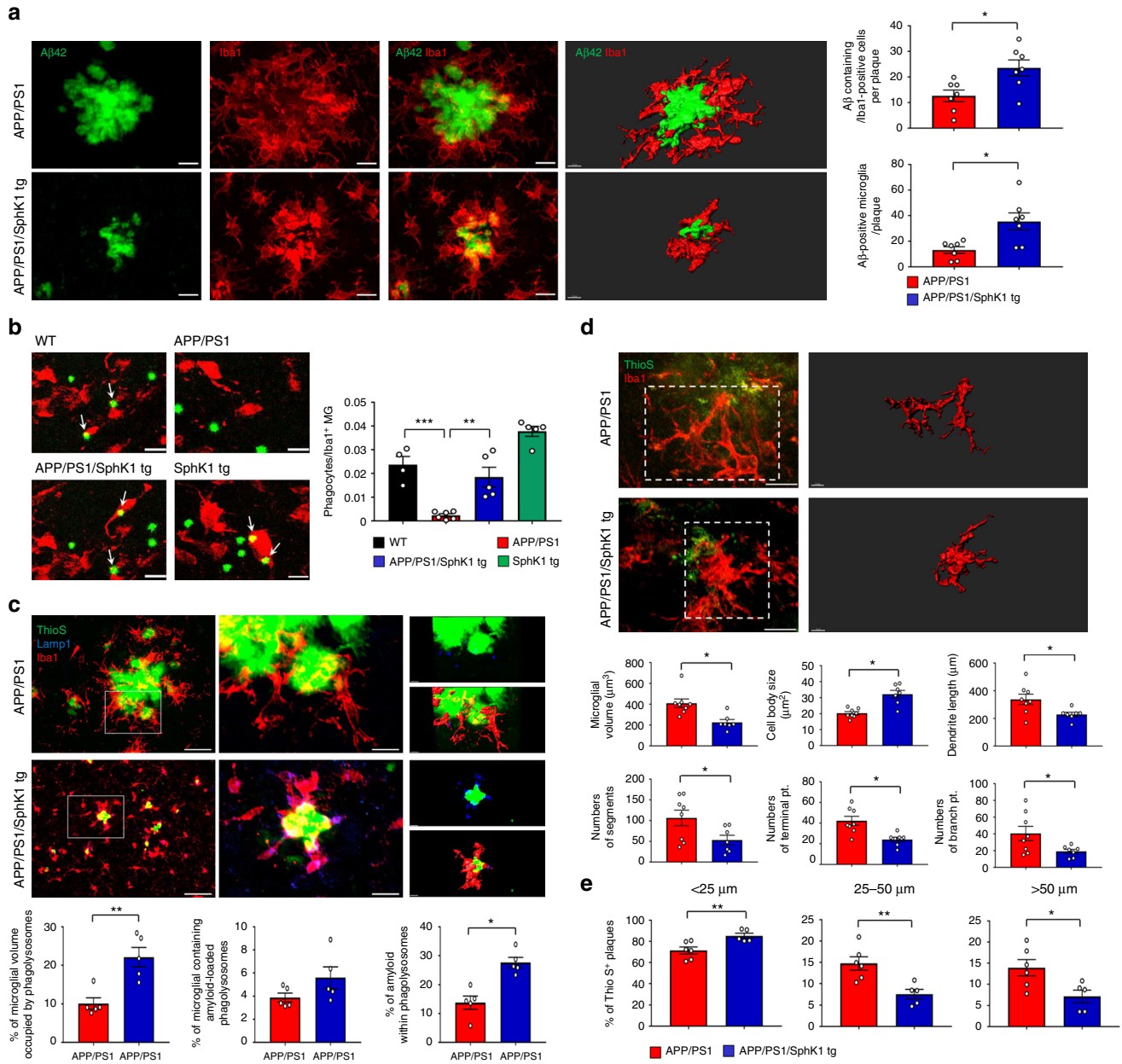

**Fig. 5** Increased SphK1 improves microglial phagocytosis. **a** Colocalization of microglia (Iba1, red) with Aβ (Aβ42, green) and quantification. Scale bars, 10 μm; 3D reconstruction from confocal image stacks scale bars, 10 μm (n = 7 per group). **b** Left, representative photomicrograph of live slice section incubated with fluorescent beads (green). Scale bar, 10 μm. White arrow point to phagocytotic microglia with fluorescent beads. Right, quantification of the number of microglial phagocytes normalized to the total number of microglia (n = 4–6 per group). **c** Up, immunofluorescence images of thio S (Aβ plaques, green) encapsulated within Lamp1+ structures (phagolysosomes, blue) in microglia (Iba1, red) present in brains of APP/PS1 and APP/PS1/SphK1 tg mice. Low-magnification scale bars, 50 μm; High-magnification scale bars, 10 μm; 3D reconstruction from confocal image stacks scale bars, 10 μm. Down, quantitation of microglial volume occupied by Lamp1+ phagolysosomes, percent of microglia containing Aβ-loaded phagolysosomes and Aβ encapsulated in phagolysosomes (n = 5 per group). **d** Morphology of microglia (Iba1, red) surrounding Aβ (ThioS, green) in cortex of APP/PS1 and APP/PS1/SphK1 tg mice. Up, high magnification (Scale bars, 10 μm) and Imaris-based three-dimensional images (Scale bars, 5 μm) of microglia surrounding Aβ. Down, Imaris-based automated quantification of microglial morphology (n = 7–8 per group). **e** Morphometric analysis of Aβ plaques in APP/PS1 and APP/PS1/SphK1 tg mice (n = 5–6 per group). Brain sections were labeled with thio S and plaques were counted and assigned to three mutually exclusive size categories based on maximum diameter: small <25 μm; medium 25–50 μm; or large >50 μm. All data analysis was performed on 9-month-old mice. **a**, **c**−**e**, Student's t test. **b** One-way analysis of variance, Tukey's post hoc test. *P < 0.05, **P < 0.01, ***P < 0.001. All error bars indicate s.e.m.

brain slices. More phagocytic microglia were found in APP/PS1/SphK1 tg mice compared to APP/PS1 mice using a fluorescent bead phagocytosis assay (Fig. 5b). To further investigate this effect, Aβ phagocytic aptitude of microglia were evaluated in vivo. APP/PS1/SphK1 tg brains showed demonstrably increased number of microglia that co-stained with lysosomes and Aβ. Importantly, phagolysosomes within microglia were increased in the cortex of APP/PS1/SphK1 tg mice compared to APP/PS1 mice. Analysis of plaque-associated microglia revealed an increased proportion of cells containing phagolysosome-encapsulated Aβ in SphK1-overexpressed APP/PS1 mice, although this trend did not reach statistical significance. Finally, the total amount of Aβ within phagolysosomes was significantly augmented in APP/PS1/SphK1 tg versus APP/PS1 brains (Fig. 5c). We also performed analysis for morphological characterization of microglia surrounding Aβ in APP/PS1 and APP/PS1/SphK1 tg mice. The phagocytic morphology (amoeboid microglial morphology) was more observed in APP/PS1/SphK1 tg mice than APP/PS1 mice (Fig. 5d). Next, we analyzed the expression of Aβ degrading enzymes, including neprilysin, matrix metallopeptidase 9, and insulin-degrading enzyme, because the immunohistochemistry cannot functionally distinguish between increased phagocytosis and impaired degradation. These enzymes remained unchanged, whereas CD36, which is known to be increased in phagocytic microglia[5], was enhanced in APP/PS1/SphK1 tg mice (Supplementary Fig. 10a, b). Microglial phagocytosis induces the reduction in the outer parts of Aβ more than in the core[4,5]. In the Aβ plaque morphometric analysis, APP/PS1/SphK1 tg mice showed significantly increased small (<25 μm) plaques. Yet, medium (25–50 μm) and large-sized (>50 μm) plaques were significantly reduced, indicating that the outer parts of the Aβ was phagocytosed by microglia (Fig. 5e). Overall, these results demonstrated that SPMs secreted by increased neuronal SphK1 enhanced the microglial Aβ phagocytic function in APP/PS1 mice.

**Neuronal SphK1 deficiency mimics AD-like pathology**. To more closely study the role of neuronal SphK1 in neurons, we crossed mice expressing cre recombinase under the control of the CamK2 promoter (CamK2-cre mice) with SphK1$^{flox/flox}$ to drive SphK1 deletion in neurons (Supplementary Fig. 11a). SphK1 mRNA expression and SphK activity were decreased in neurons of CamK2-cre;SphK1$^{flox/flox}$ mice, while no significant differences were observed in microglia and astrocytes (Supplementary Fig. 11b). Similar to previous results, sphingosine and S1P levels did not differ between the groups (Supplementary Fig. 11c). SphK1 immunoreactivity also was markedly reduced in neuron of CamK2-cre;SphK1$^{flox/flox}$ mice (Supplementary Fig. 11d). These data demonstrated that CamK2-cre;SphK1$^{flox/flox}$ mice were suitable to examine whether neuronal SphK1 itself regulate an inflammatory response, including microglia dysfunction.

First, to examine if neuronal SphK1 modulates microglial phagocytosis as shown in APP/PS1 mice, we performed a phagocytosis assay. Fewer microglia from CamK2-cre;SphK1$^{flox/flox}$ mice were observed versus control mice using the phagocytosed fluorescent bead assay (Supplementary Fig. 11e). The enzymes related with degradation were not changed, but expression of CD36 was reduced in CamK2-cre;SphK1$^{flox/flox}$ mice (Supplementary Fig. 11f). Next, we injected Aβ into the cortex of CamK2-cre;SphK1$^{flox/flox}$ mice and performed the Aβ phagocytosis assay (Supplementary Fig. 12a). CamK2-cre;SphK1$^{flox/flox}$ mice exhibited a decrease of Aβ phagocytic activity, although microglia surrounding the injected Aβ did not have any difference between control and CamK2-cre;SphK1$^{flox/flox}$ mice (Supplementary Fig. 12b, c). Injected Aβ were still remained in cortex of CamK2-cre;SphK1$^{flox/flox}$ mice more than control mice,

suggesting that SphK1 deficiency in neuron interfere with amyloid uptake (Supplementary Fig. 12d, e). SphK1-mediated COX2 acetylation and 15-R-LxA4 secretion also were decreased in neurons derived from CamK2-cre;SphK1$^{flox/flox}$ mice, and were restored after 740YP treatment (Supplementary Fig. 13a, b). Off/On signals were not changed in SphK1-deficient neurons (Supplementary Fig. 13c, d). Interestingly, CamK2-cre;SphK1$^{flox/flox}$ mice exhibited abnormal inflammatory response (Supplementary Fig. 14a−c). Similar to previous results, microglia activation was initiated prior to astrocyte activation in CamK2-cre;SphK1$^{flox/flox}$ mice (Supplementary Fig. 14d, e). Synaptic health and memory function were also impaired in CamK2-cre;SphK1$^{flox/flox}$ (Supplementary Fig. 14f−i and 15). Taken together, these results supported the fact that neuronal SphK1 deficiency reduced microglial phagocytosis by downregulating 15-R-LxA4 secretion through COX2 acetylation, and evoked AD-like brain pathology including abnormal inflammation and neuronal dysfunction.

**Neuronal SphK1 is reduced in AD patient-specific neurons**. In order to assess the clinical importance of neuronal SphK1, SphK activity and SphK1 expression were measured in the cortex from AD and control human subjects. SphK levels were decreased in brain of AD patients compared with those of control subjects (Fig. 6). To explore whether the observed effects of SphK1 in the previous mouse results were paralleled by similar alterations in AD human neurons, we first established induced pluripotent stem cell (iPSC) with PS1 and ApoE4 mutations, and induced them to differentiate into neurons, as our previous studies[25,26]. SphK1 mRNA expression and activity were significantly declined in PS1 and ApoE4 iPSC-derived neurons compared with normal iPSC-derived neuron. However, SphK2 mRNA expression and lipid levels did not differ between the groups (Supplementary Fig. 16a, b). PS1 and ApoE4 iPSC-derived neurons also showed significantly lower SphK1-mediated COX2 acetylation and SPMs secretion than neurons from normal iPSC (Supplementary Fig. 16c, d). These were restored to normal ranges by 740YP treatment that activates SphK1 (Supplementary Fig. 16e, f). Our results confirmed that reduction of SPMs secretion by SphK1-mediated COX2 acetylation occurred in AD patient neurons, and activation of SphK1 back to normal levels might improve microglial function by inducing SPMs secretion.

**Discussion**
Recently, SphK has been shown as a key factor regulating the inflammatory response. SphK1 was increased in lipopolysaccharide-activated microglia and regulated expression of pro-inflammatory cytokines in microglia. In addition, it was suggested that decreasing SphK1 expression in microglia could reduce neuroinflammation[27−29]. Unlike previous studies, we found that neuronal SphK1 was a main regulator of inflammatory response in AD. SphK1 was significantly reduced in the neuron, but not in microglia on APP/PS1 mice. SphK1 tg mice exhibited widespread and high expression levels of the transgene transcript in a variety of organs and cells, including neurons and microglia of brain[30]. In contrast with elevated microglial SphK1-induced inflammation[28,29], APP/PS1/SphK1 tg mice showed decrease of inflammation although SphK1 expression was increased in microglia. These results suggest that neurons are the major population responsible for Aβ-mediated inflammation by SphK1 than microglia.

In the current study, dysfunction of microglia, including increase of pro-inflammatory marker and loss of phagocytic function, was reported in AD. The phagocytic capacity of

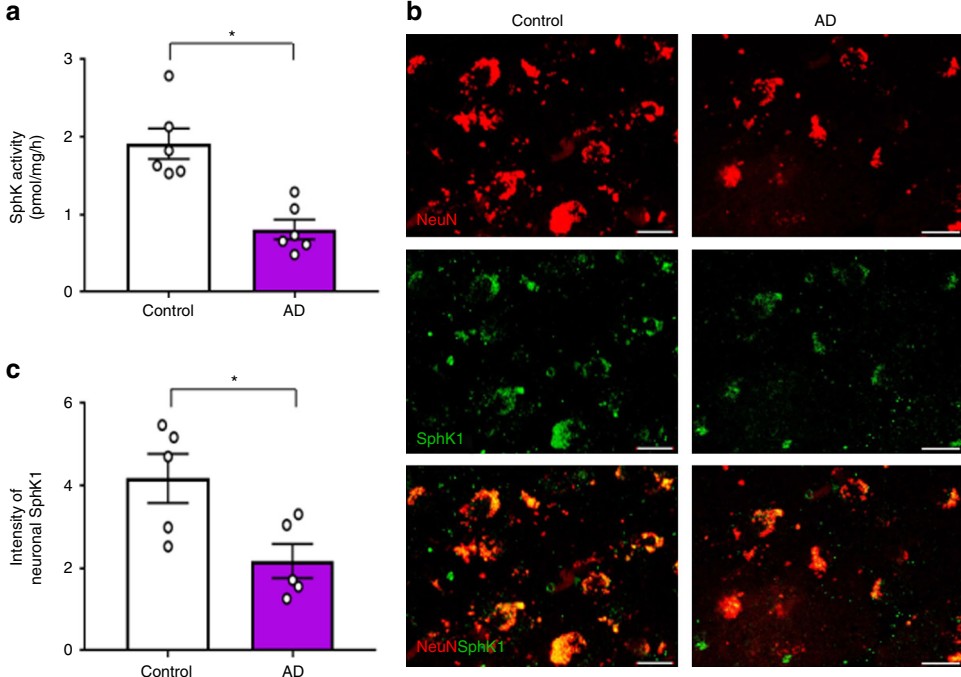

**Fig. 6** SphK1 is decreased in AD patient brains and neurons. **a** Characterization of SphK activity in cortex brain samples from AD and control human subjects ($n = 6$ per group). **b** Representative immunofluorescence images of cortex brain samples from AD and control human subjects showing SphK1 (green) merged with neuron (NeuN, red). Scale bars, 20 μm. **c** Quantification of neuronal SphK1 ($n = 5$ per group). **a**, **c** Student's $t$ test. *$P < 0.05$. All error bars indicate s.e.m.

microglia in AD can be restored, which may have important implications for future treatment of AD[31,32]. Our results showed that increase of SphK1 in APP/PS1 mice led to improved microglia function, such as increase of phagocytic ability, reduction of pro-inflammatory markers, and elevation of anti-inflammatory markers. We also found that conditional SphK1 deficiency in neurons evoked AD-like microglia dysfunction and abnormal inflammation, suggesting that neuronal SphK1 influence on microglia function and phenotype, and may be promising for treatment of AD. The brain possesses intrinsic mechanisms to modulate microglial activation through cell−cell interactions between microglia and neurons[16]. One such example is the Off/On signal expressed in neurons and we found that neuronal SphK1 did not affect this pathway. Recently, SPMs also have been shown to stimulate microglial phagocytosis and downregulate neuroinflammation[7,33]. Decreased SPMs and abnormal resolution of inflammation have been observed in patients with AD[9]. We found for the first time that SPMs secretion was declined in APP/PS1 mice and AD patient-derived neurons, and increased SphK1 in neuron restored microglia phagocytosis by elevating SPMs secretion. These findings support that neuronal SphK1 results in resolution of inflammation by regulating microglial function in AD. A previous study showed that neuron−astrocyte interactions via GABA caused memory impairment and contributed to pathology in AD[34]. Although astrogliosis as well as microgliosis also were increased in our AD mice, reduction of neuronal SphK1 was directly related to activation of microglia, but not astrocytes. These data indicate that neuron−microglia interactions are more responsive than neuron−astrocyte interactions to SphK1-mediated neuroinflammation. The differences between the previous and this study might be related to the brain regions that were examined. This study focused on the cortical region but previous studies concentrated on the hippocampus. Therefore, depending on the brain region, cell−cell interactions

might be different, a hypothesis that remains to be tested in the future.

Recent reports have also shown that the use of nonsteroidal anti-inflammatory drugs (NSAIDs) reduced the risk for the disorder and AD pathology[1,35,36]. NSAIDs generally inhibit the activity of COX1 and COX2, which were identified as the main targets of the inflammatory response. COX2 is a key player in initiating the inflammatory response by converting AA and ω-6 polyunsaturated fatty acid into pro-inflammatory prostaglandins (mainly PGE2) and triggering production of other pro-inflammatory chemokines and cytokines[18]. Because of this, a therapeutic strategy for inflammatory diseases including AD has involved inhibition of COX2. However, this is now challenged by the finding that COX2-derived SPMs possess anti-inflammatory and antioxidant properties[18]. Interestingly, production of SPMs increased in the presence of aspirin, which acetylates and inhibits COX2[18]. This indicates that, unlike selective COX2 inhibitors or other NSAIDs that do not acetylate COX2, aspirin is unique in that it not only is able to inhibit the pro-inflammatory pathway but also triggers formation of SPMs. A recent study has also reported that aspirin treatment led to positive results in an AD mouse model[37]. However, aspirin use in AD patient has indicated no effect on pathology[38]. These results might be related to the fact that aspirin has multiple functions, such as COX1 inhibition, COX2 inhibition, and COX2 acetylation. It therefore seems difficult to definitively link its use to the production of SPMs. Our study showed for the first time that COX2 acetylation was reduced in AD neurons and regulated by specifically neuronal SphK1. Moreover, SphK1 regulated COX2 acetylation-derived SPMs secretion but not affected expression of COX2 mRNA and protein. These results indicate that an SphK1-acetytransferase activator on COX2 might be a novel therapeutic approach for AD.

SphK1 and SphK2 are lipid kinases that catalyzes the conversion of sphingosine to S1P. Although these two isoenzymes have distinct physiological functions, their distinctive roles in neurons

have not been extensively investigated. We confirmed that SphK2 is more responsible for generating S1P from sphingosine than SphK1, and, for the first time, defined a new role of neuronal SphK1-acetyltransferase activity on COX2. Neuronal SphK1 showed acetyl-CoA-dependent acetyltransferase activity in the presence of [14C] acetyl-CoA, resulting in the increase of COX2 acetylation. Based on these results, we suggest the new role of neuronal SphK1 as an acetyl-CoA-dependent cytoplasmic acetyltransferase on COX2. Furthermore, our results identified the specific site of acetylation on COX2 promoted by SphK1, S565, and indicated that sphingosine is needed for this reaction to proceed. We are currently conducting further studies to investigate the in-depth mechanism for the acetyl-CoA-dependent acetyltransferase activity of SphK1 on COX2, including structural analysis of SphK1-bound acetyl-CoA and further deciphering the role of sphingosine in this reaction.

Overall, the data presented here show that decreased SphK1 levels in AD neurons worsen microglia function, resulting in failure of resolution and exacerbation of AD pathology, namely Aβ plaque deposition and cognitive impairment (Supplementary Fig. 17). The correction of neuronal SphK1 elevated SPMs secretion by inducing COX2 acetylation, and effectively blocked AD progression. These findings describe new properties and functions for SphK1, identify new mechanisms contributing to AD, and suggest a new therapeutic approach for AD and perhaps other inflammatory conditions by modulating SphK1 activity.

## Methods

**Mice.** Transgenic mouse lines overexpressing the hAPP695swe (APP) and presenilin-1M146V (PS1) mutations were originated from GlaxoSmithKline (Harlow, UK)[39] and maintained as described previously[25]. SphK1 tg mice (C57BL/6 background)[30] were bred with APP/PS1 mice to generate APP/PS1/SphK1 tg mice. Because APP/PS1 mice show sex differences in disease progression, we used only male mice. Data analysis was done in 3-, 6-, 9- or 12-month-old. SphK1−/− mice (C57BL/6 background)[40] and SphK2−/− mice (C57BL/6 background)[11] were used to detect SphK activity and lipid in neurons. CamK2-cre[41] (The Jackson Laboratory) and SphK1flox/flox mice[42] were used to delete SphK1 in neurons. Block randomization method was used to allocate the animals to experimental groups. To eliminate the bias, we were blinded in experimental progress such as data collection and data analysis. Mice were housed at a 12 h day/12 h night cycle with free access to tap water and food pellets. All protocols were approved by the Kyungpook National University Institutional Animal Care and Use Committee (IACUC).

**Cell isolation.** Adult neurons and microglia were isolated from the mice brain as previously described[43]. In brief, the cortex of WT (2, 3, 6, 9 or 12-month-old), APP/PS1 (3, 6, 9 or 12-month-old), APP/PS1/SphK1 tg (9-month-old), SphK1 tg (2 or 9-month-old), SphK1−/− (2-month-old), SphK2−/− (2-month-old) or CamK2-cre;SphK1flox/flox (2-month-old) mice was minced in Hibernate A (GIBCO)/B27 (Invitrogen) medium and dissociated using papain (Worthington) solution. After tissue trituration, cells were separated by Optiprep (Sigma-Aldrich) density gradient centrifugation. The purity of the fractionated neurons and microglia were acutely confirmed by neuronal (NeuN) and microglial (Iba1) markers and the isolated neurons and microglia were acutely analyzed for SphK mRNA levels, SphK activity, and the acetylation assay. Adult astrocytes were isolated based on earlier protocols[44]. The mice were anesthetized and perfused with Hank's Balanced Salt Solution without calcium or magnesium (HBSS) (Invitrogen). The cortex were dissected out and kept in ice-cold HBSS. After mechanical dissociation, the tissue was subjected to enzymatic dissociation using papain solution and incubated at 37 °C for 50 min on a rocker. The mix was spun at 200 × g for 15 min and the pellet was triturated to obtain a single-cell suspension. After enzymatic dissociation, cells were resuspended in 30% Percoll (GE Healthcare) and centrifuged for 10 min at 700 × g. The supernatant containing the myelin was removed, and the pelleted cells were washed with HBSS. Pelleted cells were resuspended in PBS (0.5% BSA) and stained with PE-GLAST-1 (Miltenyi Biotec, 130-095-821) and APC-CD11b (eBioscience, 17-0112-82) at 4 °C in the dark for 30 min. The GLAST-1+CD11b− cell population were sorted as astrocytes using AriaIII (BD Science).

**Primary cell culture.** To confirm the relationship of Aβ and SphK1 in neurons, cells from E18 C57BL/6 mice were prepared as previously described with minor modifications[45]. Cortices were dissected and then dissociated followed by incubation in papain for 15 min at 37 °C. Neurons were plated on poly-L-lysine-coated coverslips with neuronal culture medium, serum-free Neurobasal medium (GIBCO) containing 2% B27 supplements (GIBCO), 1 mM Glutamax supplement

(GIBCO), and 100 U/ml streptomycin/100 U/ml penicillin (GIBCO) at 37 °C in a humidified atmosphere of 5% CO2. Aβ 1-42 (10 μM, 24 h, Sigma-Aldrich), 740YP (10 μM, 24 h, Tocris Bioscience) or rapamycin (10 nM, 24 h, Cell Signaling) were exposed to neurons and cells were analyzed for PI3K signaling. Primary astrocyte cultures were prepared from C57BL/6 mice as described previously[46]. In brief, after removal of the meninges, postnatal day 7 (P7) mouse brain tissues were minced and incubated in a rocking water bath at 37 °C for 30 min in the presence of 0.25% trypsin-EDTA (Sigma-Aldrich). Enzyme-digested dissociated cells were triturated with astrocyte-specific medium (DMEM/F12 containing 10% FBS, 0.2 and 1% penicillin–streptomycin) and centrifuged at 1300 rpm for 8 min. The pellet was resuspended in DMEM/F12, passed through a 40-μm cell strainer. The filtrate was allowed pre-adherence for 30 min to remove any contamination from fibroblasts before being seeded in dishes and added astrocyte-specific medium. For astrocytes splitting, dishes were added with Ara-C and placed in a heated shaker for 6–7 h. The medium was removed from the dishes, and trypsin-EDTA was added to the dish and incubated for 5–10 min at 37 °C. After centrifugation at 1300 rpm for 8 min, the supernatant was removed and the astrocytes were maintained in culture by feeding every 1–2 weeks with astrocyte-specific medium.

**SphK activity assays.** The brains or cells were lysed in homogenization buffer containing 50 mM HEPES (GIBCO), 150 mM NaCl (Sigma-Aldrich), 0.2% Igepal (Sigma-Aldrich) and protease inhibitor (Calbiochem)[26]. We performed the enzymatic activity measurements as previously described[26] using a UPLC (Ultra performance liquid chromatography) system (Waters). Briefly, 3 μl of the samples were mixed with 3 μl of SphK (200 μM NBD-sphingosine (Molecular Probes), 100 mM of HEPES buffer, pH 7.2, 10 mM MgCl2, 200 mM Semicarbodize, 1 mM 4-deoxypyridoxin, 2 mM Dithiothreitol, 1 mM ATP and 0.2% Igepal CA-630) assay buffer and incubated at 37 °C for 1 h. The hydrolysis reactions were stopped by adding 54 μl of ethanol, and centrifuged at 13,000 rpm for 5 min. Thirty microliters of the supernatant was then transferred to a sampling glass vial and 5 μl was applied onto a UPLC system for analysis. SphK activity was followed as phosphorylation of (7-nitro-2–1,3-benzoxadiazol-4-yl)-derythro (NBD)-sphingosine (Avanti Polar Lipids) to NBD-S1P. Quantification was achieved by comparison with NBD-S1P (Avanti Polar Lipids) standards.

**Lipid extraction and sphingosine/S1P quantification.** The brains or cells were prepared as described above. To quantify the sphingosine and S1P levels, the dried lipid extract was resuspended in 0.2% Igepal CA-630. Four microliters of the lipid extracts was added into 20 μl of NDA derivatization reaction mixture (25 mM borate buffer, pH 9.0, containing 2.5 mM each of NDA and NaCN). The reaction mixture was diluted 1:3 with ethanol, incubated at 50 °C for 10 min and centrifuged (13,000 × g for 5 min). An aliquot (30 μl) of the supernatant was then transferred to a sampling glass vial and 5 μl was applied onto a UPLC system for analysis. The fluorescent sphingosine or S1P derivatives were monitored using a model 474 scanning fluorescence detector (Waters). Quantification of the sphingosine and S1P peaks were calculated from sphingosine and S1P standard calibration curves using the Waters Millennium software.

**Histological analysis.** Thioflavin S staining was carried out according to previously described procedures[47]. We used SphK1 (rabbit, 1:50, Abgent, 8877), NeuN (mouse, 1:200, Millipore, MAB 377), 6E10 (mouse, 1:100, Signet, SIG39300), anti-20G10 (mouse, 1:1000, provided by D.R. Howlett, GlaxoSmithKline, Harlow, Essex, UK) for Aβ42, anti-G30 (rabbit, 1:1000, provided by D.R. Howlett) for Aβ40, SMA (mouse, 1:400, Sigma-Aldrich, A2547), AT8 (mouse, 1:500, Thermo Fisher Scientific, MN1020), MAP2 (chicken, 1:2000, Abcam, ab5392), Synaptophysin (rabbit, 1:100, Abcam, ab32127), Synapsin1 (rabbit, 1:500, Synaptic systems, 106 103), PSD95 (mouse, 1:100, Millipore, MAB1596), Iba1 (rabbit, 1:500, Wako, 019-19941), GFAP (rabbit, 1:500, Dako, N1506), S100 (rabbit, 1:200, Dako, Z0311), SOX2 (mouse, 1:100, R&D Systems, MAB2018), COX2 (rabbit, 1:50, Santacruz, SC7951), LOX-15 (rabbit, 1:100, Cayman chemical, 160704) and Lamp1 (mouse, 1:200, Abcam, ab24170). The sections were analyzed with a laser-scanning confocal microscope (FV1000; Olympus) or with a BX51 microscope (Olympus). MetaMorph software (Molecular Devices) was used for quantification. Three-dimensional reconstruction of microglia was recorded and analyzed using IMARIS software (Bitplane)[48].

**Western blotting.** Samples were lysed in RIPA buffer (Cell Signaling Technologies), then subjected to SDS–PAGE and transferred to a nitrocellulose membrane. Membranes were blocked with 5% milk, incubated with primary antibody and then incubated with the appropriate horseradish peroxidase-conjugated secondary antibody[25,49]. Primary antibodies to the following proteins were used: SphK1 (rabbit, 1:1000, Abgent, 8877), Synaptophysin (rabbit, 1:2000, Abcam, ab32127), MAP2 (chicken, 1:10,000, Abcam, ab5392), Synapsin1 (rabbit, 1:1000, Synaptic systems, 106 103), PSD95 (mouse, 1:1000, Millipore, MAB1596), Caspase 3 (mouse, 1:200, NOVUS, 31A1067), BACE-1 (mouse, 1:1000, Millipore, MAB5308), LC3 (rabbit, 1:1000, Cell Signaling Technology, 4108S), Beclin-1 (rabbit, 1:1000, Cell Signaling Technology, 3738S), rab5 (rabbit, 1:1000, Cell Signaling Technology, 3547S), rab7 (rabbit, 1:1000, Cell Signaling Technology, 9367S), 6E10 (mouse, 1:500, Signet, SIG39300), PI3K (rabbit, 1:1000, Cell Signaling Technology, 5569), p-AKT (rabbit, 1:1000, Cell Signaling Technology, 4060), p-mTOR (rabbit, 1:1000,

Cell Signaling Technology, 5536), p-ERK (rabbit, 1:1000, Cell Signaling Technology, 9101), CD36 (rabbit, 1:500, Novus Biologicals, NB400-144) and β-actin (1:1000, Santa Cruz, Biotechnology, Inc., SC-1615). We performed densitometric quantification using the ImageJ software (National Institutes of Health). Images have been cropped for presentation. Full size images are presented in Supplementary Fig. 18.

**ELISA**. For measurement of Aβ40 and Aβ42, we used commercially available ELISA kits (MyBioSource). Cortex of mice was homogenized in buffer containing 0.02 M guanidine. ELISA was then performed for Aβ40 and Aβ42 according to the manufacturer's instructions. For measurement of LxA4, RvE1 and RvD1 levels were assayed by using a mouse and human LxA4, RvE1, RvD1 kit (MyBioSource) according to the manufacturer's instructions.

**SphK1 siRNA treatment**. *SphK1* siRNA (5′-GGUACGAGCAGGUGACUAA-3′; 5′- UAGCAAGCCUGCGCAUCUA-3′; 5′-GGAGAUUCGUUUCACAGUG-3′; 5′- AGGCAGAGAUAACCUUUAA-3′, Dharmacon SMART pool) and scrambled sequence siRNA control (5′-UGGUUUACAUGUCGACUAA-3′; 5′-UGGUUUA-CAUGUUGUGUGA-3′; 5′-UGGUUUACAUGUUUUCUGA-3′; 5′-UGGUUUA-CAUGUUUUCCUA-3′, Dharmacon SMART pool) were treated in neurons from E18 C57BL/6 mice for 48 h. Cells were collected and analyzed for acetylation assay.

**Acetylation assay**. Acetylation assay was performed according to previously described procedures with minor modifications[50]. Neurons were sonicated and resuspended in acetylation buffer (50 mM Tris/HCl; pH 8.0, 0.1 mM EDTA, 1 mM DTT, 10% glycerol). The supernatants were incubated with either [acetyl-$^{14}$C] aspirin (2 μCi, ARC UK Ltd,) or [$^{14}$C] acetyl-coa (2 μCi, PerkinElmer) (2 h, 37 °C). Also, COX2 protein (10 μg, LSBio, LS-G21094) was incubated with SphK1 (0.05 mM, Cayman Chemical, 10348) and sphingosine (100 μM, Sigma-Aldrich) or SphK1 only in the presence of [$^{14}$C] acetyl-coa (2 h, 37 °C). The reaction was terminated by placing the reaction tubes in ice for 5 min. For immunoprecipitating acetylated COX2, the sample was incubated with COX2 antibody (rabbit, Santa-cruz, SC7951) (24 h, 4 °C) and precipitated by adding 50 μl protein-A sepharose (GH Healthcare). Acetylated COX2 proteins bound to bead were analyzed using a Tri-Carb 3110TR liquid scintillation counter (Perkin-Elmer).

**LC-MS/MS**. To confirm the relationship of SphK1 and SPMs secretion in neurons, neurons were isolated from WT, APP/PS1, APP/PS1/SphK1 tg and SphK1 tg mice at 9 months. The neurons were sonicated and resuspended in neuronal culture medium. The supernatants were incubated with 2.5 mM acetyl-coa (Sigma) (24 h, 37 °C). Also, CM were harvested from primary neuron treated with *SphK1* siRNA or control siRNA. A 200 μl aliquot of each cell lysate or CM was added to a 100 μl aliquot of 100 pg/ml of 15-S-LxA4-d5 (internal standard, Cayman Chemical) solution, 100 μl of 1% formic acid solution, and 600 μl of water. After mixing thoroughly, 4 ml of ethyl acetate was added. After vortexing and centrifuging (13,200 rpm) 10 min each, the mixture was frozen in a deep freezer for 2 h. The organic supernatant was separated and dried under nitrogen stream. The residue was reconstituted with 60% acetonitrile solution injected into the LC-MS/MS system.

The analysis of the 15-R-LxA4 concentration was performed using an Agilent 6470 Triple Quad LC-MS/MS system (Agilent, Wilmington, DE, USA) coupled to an Agilent 1290 HPLC system. Chromatographic separation was achieved using a Lux Amylose-2 (3 μm, 2.0 mm i.d. × 150 mm, Phenomenex), connected to a Synergi Polar RP (4 μm, 2.0 mm i.d. × 150 mm, Phenomenex). Analytes were eluted with a mobile phase consisting of water:formic acid (100:0.1 v/v) (phase A) and acetonitrile:formic acid (100:0.1 v/v) (phase B) in gradient elution mode for 21 min at a flow rate of 0.2 ml/min. The elution gradient was as follows: from 0 to 0.5 min the content of phase A was 50% and within 1 min, the content of phase A was decreased to 10% and maintained for 8.5 min. Within 0.5 min the content of phase A was increased back to 50% and maintained for 11.5 min. Quantification was carried out using multiple reaction monitoring at m/z 351.1 → 115.1 for 15-R-LxA4 and 15-S-LxA4 and m/z 356.1 → 115.1 for 15-S-LxA4-d5 (internal standard) in negative ionization mode and collision energy of 15 eV. In this study, the peak areas for all components were automatically integrated using MassHunter B 06.00. The lower limit of quantification (LLOQ) was determined to be 10 pg/ml and linearity was observed in the standard range of 10–200 pg/ml.

To identify acetylation site of COX2, COX2 enzymes were immediately precipitated with trichloroacetic acid (Merck) and dried. The dried extract was resuspended in 10 μl of 5 M urea solution and incubated with 1 μg sequencing-grade modified porcine trypsin (Promega) in 0.1 M ammonium bicarbonate buffer at 37 °C for 16 h. Then the sample was treated with 1 M DTT (GE Healthcare) for 1 h at room temperature, followed by alkylation with 1 M iodoacetamide (Sigma) for 1 h. For sequencing, the protein samples were loaded onto ZORBAX 300SB-C18 column (3.5 μm, 1.0 mm i.d. × 150 mm, Agilent). The column was placed in-line with an UltiMate 3000 system (Dionex, US) and a splitter system was used to achieve a flow rate of 100 μl/min. Analytes were eluted with a mobile phase consisting of water:formic acid (100:0.2 v/v) (phase A) and acetonitrile:formic acid (100:0.2 v/v) (phase B) in gradient elution mode for 77 min. Eluted peptides were directly electrosprayed into and MicroQ-TOF III mass spectrometer (Bruker Daltonics, 255748, Germany) by applying 4.5 kV of capillary voltage and a normalized collision energy of 7 eV. The peptides were verified with BioTools 3.2 SR5 (Bruker Daltonics).

**COX2 point mutation**. Point mutations in COX2 (S565A) was generated using the In-Fusion Cloning Kits (Clontech) and human *COX2* gene ORF cDNA clone expression plasmid (Catalog number: HG12036-NH, Sino Biological Inc), according to the manufacturer's instructions, and the following primer and its reverse-complement for point mutation were used: Forward 5′-CTTTACTGCTTTCAGTGTTCCAGATCCAGAGC-3′; Reverse 5′-CTGAAAGCAGTAAAGGGACAGCCCTTCACG-3′. The COX2 WT plasmids and *COX S565A* plasmids were transformed in the One shot TOP 10 competent cell (Invitrogen). The bacteria were grown in Luria-Bertani (LB) plate supplemented with 50 μg/ml kanamycin at 37 °C. The bacterial cell pellets were resuspended in lysis buffer containing 50 mM Tris (PH 8.0), 500 mM NaCl (Sigma-Aldrich), complete EDTA-free protease inhibitor (Roche), DNase I (Roche) and 0.25 mM DTT before being lysed by sonication and clarified by centrifugation at 75,600 × g. Proteins were purified from the soluble fraction using nickel-NTA agarose (QIAGEN) and acetylation assay was performed.

**Kinetic analysis of acetyltransferase activity**. Acetyl-CoA binding activity of SphK1 was analyzed by filter binding assay[22]. Briefly, SphK1 enzymes (1 mU, Cayman Chemical) were resuspended in 100 μl reaction buffer (20 mM Hepes; pH 7.4, 50 mM NaCl, 10 mM MgCl$_2$, 1 mM EGTA, and 0.02% Triton-X100). Reaction was initiated by adding [$^{3}$H] acetyl-CoA (1–200 μM (0.1–20 μCi), PerkinElmer) and incubated for 1 h at 37 °C. The reaction was terminated by adding 100 μl of ice-cold reaction buffer containing 5 mM cold acetyl-CoA and immediately filtered through P30 filtermat (PerkinElmer), followed by five times washing. Incorporated [$^{3}$H] acetyl-CoA to SphK1 was analyzed using a Micro Beta 2 liquid scintillation counter. The binding velocity of [$^{3}$H] acetyl-CoA to SphK1 was plotted to the acetyl-CoA concentration and the $K_{cat}$ and $K_M$ values were calculated from the Michaelis−Menten analysis using Sigma-plot program (ver 10.0, Systat Software Inc.). $K_{cat}$ and $K_M$ indicated the catalytic constant and the Michaelis−Menten constant, respectively. Dissociation constant for incorporated [$^{3}$H] acetyl-CoA to SphK1 was measured by equilibrium dialysis[51]. The incorporated [$^{3}$H] acetyl-CoA to SphK1 was purified by rapid equilibrium dialysis kit (ThermoFisher) and unbound acetyl-CoA was less than 10%. The [$^{3}$H] acetyl-CoA and SphK1 complex was placed in 100 μl reaction buffer in the presence of 0, 5, 25, and 100 μM free acetyl-CoA inside the dialysis chamber and 5 ml of fresh reaction buffer was placed outside the chamber. The released free [$^{3}$H] acetyl-CoA outside the chamber was measured every 30 min for 3 h. The dissociation rate was plotted versus inhibitor concentration and dissociation constant ($K_D$) was calculated from the Michaelis−Menten analysis using Sigma-plot program (ver 10.0, Systat Software Inc.)[52].

**Phagocytosis assay**. The phagocytic activity of adult microglia in the cortex of live brain slices was analyzed following the protocol[6]. Brains from WT, APP/PS1, APP/PS1/SphK1 tg, SphK1 tg and CamK2-cre;SphK1$^{flox/flox}$ mice were washed in carbogen-saturated (95% O$_2$ and 5% CO$_2$) artificial cerebrospinal fluid (ACSF) containing (in mM): NaCl 126; KCl 2.5; MgSO$_4$ 1.3; CaCl$_2$ 2.5; NaH$_2$PO$_4$ 1.25; NaHCO$_3$ 26; and D-glucose 10; pH 7.4 (all from Sigma). Coronal slices (130 μm) were prepared using a vibratome (Leica Biosystems, Nussloch, Germany) at 4 °C and allowed to rest in ACSF buffer at room temperature for 1 h before incubation with fluorescent carboxylated microspheres (1 μm diameter, FluoSpheres, Invitrogen, 1:2000) in PBS (Gibco) containing 4.5 g/l D-glucose (Sigma) for 1 h at 37 °C. The slices were washed and fixed with 4% PFA. To visualize microglia, slices were permeabilized (2% Triton X-100, 2% BSA and 10% normal goat serum in PBS) and incubated with Iba1 (rabbit, 1:500, Wako), followed by donkey anti-rabbit Alexa 594 (1:1000, Invitrogen). The sections were analyzed with a laser-scanning confocal microscope (FV1000; Olympus). MetaMorph software (Molecular Devices) was used for quantification.

**Aβ cortical injection**. CamK2-cre;SphK1$^{flox/flox}$ mice (2-month-old) were injected with Aβ (10 μM) at cortex of brain. After anesthesia with a combination of 100 mg/kg ketamine and 10 mg/kg xylazine, a stainless steel cannula was implanted in the animal's cortex using a stereotaxic frame (David Kopf Instrument, Tujunga, CA). The guide cannula was fixed in the cortex brain region according to the following coordinates: 1.8 mm posterior to the bregma, 3 mm bilateral to the midline, and 1.2 mm ventral to the skull surface. The guide cannula allows a stable aperture in the brain by indicating where the treatments can be administrated. The injection of aggregated Aβ was made bilaterally into the cortex using an injection cannula connected to a microsyringe (Hamilton). The volume of injection was 1 μl of 10 μM aggregated Aβ or 1 μl of PBS.

**Behavioral studies**. We performed behavioral studies to assess spatial learning and memory in the Morris water maze as previously described[25,47]. Animals were given four trials per day for 10 days to learn the task. At day 11, animals were given a probe trial in which the platform was removed. Fear conditioning was conducted as in previously described techniques[25]. On the conditioning day, mice were individually placed into the conditioning chamber. After a 60-s exploratory period, a tone (10 kHz, 70 dB) was delivered for 10 s; this served as the conditioned stimulus

(CS). The CS co-terminated with the unconditioned stimulus (US), a scrambled electrical footshock (0.3 mA, 1 s). The CS−US pairing was delivered twice at a 20-s intertrial interval. On day 2, each mouse was placed in the fear-conditioning chamber containing the same exact context, but with no administration of a CS or foot shock. Freezing was analyzed for 5 min. On day 3, a mouse was placed in a test chamber that was different from the conditioning chamber. After a 60-s exploratory period, the tone was presented for 60 s without the footshock. The rate of freezing response of mice was used to measure the fear memory. The open field test was used for locomotion and anxious behaviors. The open field box consisted of a square box. Each animal was placed in the box for 10 min. Overall activity in the box was measured, and the amount of time and distance traveled in the center arena was noted. After each trial, the test chambers were cleaned with a damp towel and distilled water followed by 70% alcohol.

**Human brain samples**. Neuropathological processing of control and AD human brain samples followed the procedures previously established for the Boston University Alzheimer's Disease Center. Frontal cortical regions were used for experiments. Detailed information of brain tissues is described in Supplementary Table 1. In all cases where AD was diagnosed at autopsy, AD was stated as the cause of death. AD subjects had no evidence of other neurological disease based on neuropathological examination. Next of kin provided informed consent for participation and brain donation. This study was reviewed by the Boston University School of Medicine Institutional Review Board (Protocol H-28974), and was approved as exempt because the study involves only tissue collected from postmortem, and consequently not classified as human subjects.

**Preparation of human iPSCs-derived neurons**. PS1 and ApoE4-iPSCs were established from the patient's skin fibroblasts (Coriell Institute) as previously described[25]. On day 1, the VSV-G pseudotyped retroviral vector system carrying OCT4, SOX2, KLF4, and c-Myc was added to fibroblast cultures. On day 2, cells were subjected to the same transduction procedures and harvested 24 h later. Transduced cells were replated on mouse embryonic fibroblast (MEF) layers in 100 mm$^2$ dishes containing the fibroblast medium. On the next day, the medium was changed to complete ES medium with 0.5 mM valproic acid (Sigma-Aldrich), and thereafter changed every other day. After 20 days, ES-like colonies appeared and were picked up to be reseeded on new MEF feeder cells. Cloned ES-like colonies were subjected to further analysis. Established PS1 and ApoE4-iPSCs were tested to confirm absence of mycoplasma contamination using MycoAlert PLUS Mycoplasma detection kit (Lonza, LT07). Normal iPSC line (HPS0063) was obtained from the RIKEN Bioresource Center[25]. To neural differentiation of human iPSCs[25], iPSC colonies were detached from feeder layers and cultured in suspension as embryonic body for 30 days in bacteriological dishes. Embryonic bodies were then enzymatically dissociated into single cells and the dissociated cells cultured in suspension in serum-free media hormone mix media for 10–14 days to allow the formation of neurospheres. Neurospheres were passaged repeatedly by dissociation into single cells followed by culture in the same manner. For terminal differentiation, dissociated neurospheres were allowed to adhere to poly-ʟ-ornithine- and laminin-coated coverslips and cultured for 10 days.

**RNA isolation and real-time PCR analysis**. RNA was extracted from the brain homogenates and cell lysates using the RNeasy Lipid Tissue Mini kit and RNeasy Plus Mini kit (QIAGEN) according to the manufacturer's instructions. cDNA was synthesized from 5 μg of total RNA using a commercially available kit (Takara Bio Inc.). Quantitative real-time PCR was performed using a Corbett research RG-6000 real-time PCR instrument. Primers are described in Supplementary Table 2.

**Statistical analysis**. Sample sizes were determined by G-Power software. Comparisons between two groups were performed with Student's t test. In cases where more than two groups were compared to each other, a one-way analysis of variance was used, followed by Tukey's HSD test. All statistical analyses were performed using SPSS statistical software. *$P < 0.05$, **$P < 0.01$, ***$P < 0.001$ were considered to be significant.

**Data availability**. We declare that the data supporting the findings of this study are available within the article and its Supplementary Information files and from the authors upon request.

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

## Acknowledgements

We particularly would like to thank Dr. Richard L. Proia provided SphK1 and SphK2 knockout mice. This work was supported by the Basic Science Research Program (2015R1A2A1A01004779, 2017R1A2A1A17069686) of the National Research Foundation (NRF) of Korea funded by the Ministry of Science, ICT & Future Planning (2017R1A4A1015652), Republic of Korea. This research was also supported by a grant of the Korea Health Technology R&D Project through the Korea Health Industry Development Institute (KHIDI), funded by the Ministry of Health & Welfare, Republic of Korea (HI16C2131).

## Author contributions

J.Y.L. designed and performed experiments and wrote the paper. S.H.H., M.H.P., B.B., I.-S.S., M.-K.C., S.H.K., and X.H. performed experiments and analyzed data. H.R. performed human tissue experiments. E.H.S., J.-S.B. and H.K.J. interpreted the data and reviewed the paper. J.-S.B. and H.K.J. designed the study and wrote the paper. Y.T. generated and provided SphK1 tg mice. All authors discussed results and commented on the manuscript.

## Additional information

**Competing interests:** The authors declare no competing interests.

