## [Peer Review File(PDF 1999 kb) · Nature Communications]

Reviewers' comments:

Reviewer #1 (Remarks to the Author):

This is an amazing study that provides a novel pathogenic mechanism by which Amyloid-beta (Ab) accumulation in the brain leads to microglia dysfunction. The authors show that Ab reduces Sphingosine kinase 1 (SphK1) expression in neurons due to reduced activation of the PI3K pathway. They go on to demonstrate that SphK1 has an acetylase activity which impairs the ability of COX2 to generate prostaglandins, while facilitating the synthesis of SPM. SPM released by neurons activate phagocytic activity of microglia and reduce their inflammatory function. Thus, in a model of AD neurons, release of SPM is impaired and microglia is dysfunctional. However, SPM release are reconstituted in SphK1 transgenic mice, leading to amelioration of microglial function and Ab pathology. SphK1 knockout mice show similar defects in acetylation of COX2 and generation of SPM. Brain specimens of patients with AD and iPSC show similar reductions in SphK1 function.

I find this paper extraordinary for the amount and the quality of the experiments presented, which range from basic biochemistry, to mouse studies to human studies.

Experiments are convincing, the paper is clearly written. I see no weaknesses. I congratulate the authors for this outstanding study

Reviewer #2 (Remarks to the Author):

The current manuscript by Lee and co-workers provides evidence for the reduction of SphK1 in an AD mouse model, patient iPSC-derived neurons and postmortem brains. The authors identify a novel role of neuronal SphK1 as an acetyltransferase of COX2 and a regulator of neuroinflammation and microglial function. This regulatory link is established through secretion of SPMs that help in resolving overt neuroinflammation and maintaining microglial homeostasis. In AD, reduced SphK1 signal correlates with impaired production of SPMs, prolonged inflammation and microglial dysfunction. This manuscript combines genetic and pharmacological approaches and provides solid evidence in support of beneficial role of enhancing neuronal SphK1 and COX2 acetylation to stimulate SPMs secretion and thus control inflammation and microglial function. In addition, this work nicely illustrates the importance of dissecting neuronal and microglia interactions in elucidating pathomechanisms of AD.

This study would benefit from following major/minor recommendations:

Major recommendations:

1) The amount of data shown in the manuscript is overwhelming. It would help the reader to focus on relevant data for the manuscript "take-home message" and include more details with all necessary controls while omitting some parts which seem less conclusive.

2) The authors indicate (page 32):

"The purity of the fractionated neurons and microglia were confirmed by neuronal (MAP2) and microglial (Iba1) markers..".

This is an important control and should be shown as supplementary data for all cell

populations (neurons, microglia and astrocytes).

3) SphK1 immunofluorescence analysis (Fig 1e) shows an increase in neuronal SphK1 signal in APP/PS1/SphK1 tg and SphK1 tg mice. Although a clear difference between WT and APP/PS1 can be observed, an increase between WT and APP/PS1/SphK1 tg and SphK1 tg is more prominent in quantification analysis (Fig 1f) than reflected on the representative images. It would be good to provide western-blot analysis for SphK1 levels in the brain, ideally in different cellular populations.

4) I would suggest including Suppl Fig 2c as part of the Fig 2 to pinpoint the influence of SphK1 on the reduction of amyloid plaque burden and correlate this effect with microglial morphology. A β immunofluorescence analysis of amyloid plaques (for example using 6E10 antibody) should be included. High magnification images of microglia around amyloid plaques should also be presented.

5) A decrease in levels of synaptophysin, MAP2, synapsin and PSD95 in APP/PS1 mice (Suppl Fig 2h-k) is surprisingly strong. This should be re-evaluated and discussed in the light of previous reports that show no neuronal loss in the APP/PS1 mouse line (Howlett et al., Brain Research, 2004). In addition, synaptophysin staining pattern in WT animals is unusual and completely different in Suppl Fig 2h and Suppl Fig 15f. Similarly, caspase 3 staining (Suppl Fig 4a) also displays an unusual staining pattern that rather resembles plaque-like structures in APP/PS1 mice. I would recommend evaluating the levels of synaptic proteins, MAP2 and caspase 3 by western-blotting. Of note, synaptophysin, MAP2, synapsin or PSD95 quantifications should not be indicated as burden.

6) The authors should comment on visible differences in the response of microglia to A β treated WT neurons in Suppl Fig 6a and c. If such experimental variations are to be observed it would be hard to judge the protective effect of SphK1 overexpression.

7) Morphological characterization of microglia (Suppl Fig 7b) would benefit from additional analysis such as microglial number per plaque volume and microglial volume (or soma size). The authors stated on page 19 of the manuscript that microglial recruitment to amyloid plaques was analyzed in Fig. 5a. However, the figure itself refers to A β containing/Iba1-positive cells per plaque. From the presented analysis (Fig 5) it is hard to judge whether microglia really took up A β . As this result is of importance for the major conclusion on phagocytic activity, it is necessary to show high magnification images, best combined with 3D rendering analysis, to strengthen this result. Similarly, immunofluorescent images of a phagocytosis assay presented in Fig 5b are very difficult to judge. Above suggested morphological characterization of microglia could help in correlating morphological changes with phagocytic capacity as amoeboid microglial morphology is indicative of phagocytic microglia.

8) Suppl Fig 10b shows effects of SphK1 siRNA in neurons on secretion of various SPMs. However, the authors stated before (page 15) that only 15-R-LxA4 can be detected in neurons by LC-MS/MS:

"Interestingly, only 15-RLxA4, one of COX2 acetylation-derived products, were found in

neurons”.

9) The authors analyzed A β phagocytosis in CamK2-cre+/-;SphK1flox/flox mice after A β cortical injection (Suppl Fig 13). It is unclear what do the authors mean by Iba1-positive cells per plaque (Suppl Fig 13b) or area occupied by plaques (Suppl Fig 13d) as A β injection results in inducing amyloid plaque pathology only when injected in APP/PS1 background. Immunofluorescent images from the day of the injection (0) and analysis (7) should be shown including immunostaining for amyloid and microglia.

10) Why do the numbers of phagocytes/Iba1-positive cells in control mice differ so much in Figures 5b and Suppl Fig 12e? Similar is also observed when comparing C14 signal in Suppl Figs 11a and 17c with Suppl Fig 14a.

Minor recommendations:

1) Consider including a more general reference for microglial dysfunction in AD (page 3). Additional reference is recommended for the evidence that NSAIDs may reduce risk of AD pathology (page 28).

2) I would avoid using the M1 and M2 terminology throughout the manuscript as this classification has not proven to be useful.

3) It would be helpful to indicate SphK mRNA (fold change) and SphK activity on the figure axes such as Fig. 1a to distinguish better between the two experimental paradigms.

4) The authors state on page 6:

“However, this activity was slightly decreased in cells derived from SphK2-/- mice”. This decrease is obvious and significant and slightly may not be the best description.

5) Introduction of the SphK1 tg and APP/PS/SphK1 tg mice (page 6) should come before as it is part of the Fig 1a. Consider introducing SphK1 phenotypes in all lines first and then showing the lipid analysis data. This would help the reader to keep the track.

6) Consider modifying the following sentence (page 9) as IL-10 behavior is different compared to the rest of the anti inflammatory markers and is indeed decreased in APP/PS1/SphK1 mice compared to APP/PS1 mice.

“In addition, APP/PS1/SphK1 tg mice showed a decrease of pro-inflammatory M1 markers and immunoregulatory cytokines, and an increase of anti-inflammatory M2 markers compared with APP/PS1 mice”.

The same applies to the description of Suppl Fig 7a on page 12.

7) SphK1 levels were analyzed in both neurons (Suppl Fig 5b) and microglia (Suppl Fig 5c), but microglial data are not described in the text (page 10). Suppl Fig 5f is showing SphK1 mRNA levels in neurons, but not in astrocytes or microglia while the activity has been analyzed in all cell types. The authors should be more consistent through the manuscript. For the reader it is sometimes hard to follow why certain cellular populations have been included and others not (additional example: Suppl Fig 1a).

8) Suppl Fig 5i is showing SphK1 mRNA levels in A β treated neurons and not as indicated in the text in AD brains (page 11).

9) The following sentence (page 13) is not informative:

"AD-like pathology was also improved in 12-mo-old APP/PS1/SphK1 tg mice (data not shown), suggesting that the increase of SphK1 protects disease progression even at older ages".

10) I would highly recommend streamlining and shortening the discussion.

18) It has been stated on page 27:

"It is widely accepted that microglia degenerate during AD progression and shift to an M1 activation state, with loss of phagocytic function".

Please, refer to minor recommendations/comment 2. In addition, microglial degeneration in AD has not been proven yet. This point should be discussed in the light of recent publications demonstrating that phagocytic capacity of AD microglia can be restored which should not be the case if overt degeneration of microglia would occur.

19) Additional reference could be added to statement on page 27:

"Recently, SPMs also have been shown to stimulate microglial phagocytosis and down-regulate neuroinflammation".

Manuscript by Zhu et al., Mol Neurobiol, 2016 is addressing the role of SPMs in inflammation resolution pathway in AD. However, although they also provide evidence for important regulatory of SPMs is microglial phagocytosis, this beneficial effect was not exerted by LXA4 SPM.

Reviewer #3 (Remarks to the Author):

Ju Youn Lee and co-authors present a comprehensive set of data supporting a novel and disease relevant role for neuronal SphK1 in the regulation of microglial phagocytosis by modulating COX2-mediated "specialized proresolving mediators" (SPMs) secretion. The authors show that the underlying mechanism involves a SphK1 acetyltransferase activity towards COX2 and identified Ser565 as the targeted residue. Interestingly, the presented data show that neuronal SphK1 is reduced in Alzheimer's disease (AD) patient brains/cells and this reduction lowers acetylated-COX2 levels and leads to decreased SPMs secretion. In contrast, SphK1 activation by 740YP treatment induces SPMs secretion. Therefore, the authors conclude that neuronal SphK1 acetyltransferase activity may contribute to AD pathology by impairing microglial phagocytosis and resolution of inflammation, and propose that activation of neuronal SphK1 acetyltransferase on COX2 as a potential novel therapeutic strategy in AD.

The study is highly relevant for the AD field and as discussed, may open novel therapeutic avenues. Furthermore, the elucidated mechanisms may have a broader significance in other

CNS disorders.

Major comments:

1. "APP/PS1/SphK1 tg mice showed a significant reduction of microglia and astrocytes compared to APP/PS1 mice (Fig. 2a, b)." However, the authors also show that "microglial recruitment was increased in APP/PS1/SphK1 tg mice compared APP/PS1 mice (Fig. 5a)." and "More phagocytic microglia were found in APP/PS1/SphK1 tg mice compared APP/PS1 mice (Fig. 5b)."

How do the authors explain the differential effects on SphK1 (tg mice) on the size of the microglia population and phenotype.

2. Neuronal SphK1 acetyltransferase activity depends on the presence of sphingosine and the authors suggest that acetylation may occur through a sphingosine intermediate. the authors should evaluate how sphingosine changes the kinetic (K_m , k_{cat}) parameters of SphK1. This is relevant due to the potential therapeutic value of this activity.

Minor comments:

I suggest to merge and shorten the first two sections in the results and to improve clarity figure legends should be revised.

Reviewer #1 (Remarks to the Author)

This is an amazing study that provides a novel pathogenic mechanism by which Amyloid-beta (Ab) accumulation in the brain leads to microglia dysfunction. The authors show that Ab reduces Sphingosine kinase 1 (SphK1) expression in neurons due to reduced activation of the PI3K pathway. They go on to demonstrate that SphK1 has an acetylase activity which impairs the ability of COX2 to generate prostaglandins, while facilitating the synthesis of SPM. SPM released by neurons activate phagocytic activity of microglia and reduce their inflammatory function. Thus, in a model of AD neurons, release of SPM is impaired and microglia is dysfunctional. However, SPM release are reconstituted in SphK1 transgenic mice, leading to amelioration of microglial function and Ab pathology. SphK1 knockout mice show similar defects in acetylation of COX2 and generation of SPM. Brain specimens of patients with AD and iPS show similar reductions in SphK1 function.

I find this paper extraordinary for the amount and the quality of the experiments presented, which range from basic biochemistry, to mouse studies to human studies.

Experiments are convincing, the paper is clearly written. I see no weaknesses. I congratulate the authors for this outstanding study

We thank the reviewer for these positive comments concerning our manuscript.

Reviewer #2 (Remarks to the Author)

The current manuscript by Lee and co-workers provides evidence for the reduction of SphK1 in an AD mouse model, patient iPSC-derived neurons and postmortem brains. The authors identify a novel role of neuronal SphK1 as an acetyltransferase of COX2 and a regulator of neuroinflammation and microglial function. This regulatory link is established through secretion of SPMs that help in resolving overt neuroinflammation and maintaining microglial homeostasis. In AD, reduced SphK1 signal correlates with impaired production of SPMs, prolonged inflammation and microglial dysfunction. This manuscript combines genetic and pharmacological approaches and provides solid evidence in support of beneficial role of enhancing neuronal SphK1 and COX2 acetylation to stimulate SPMs secretion and thus control inflammation and microglial function. In addition, this work nicely illustrates the importance of dissecting neuronal and microglia interactions in elucidating pathomechanisms of AD.

This study would benefit from following major/minor recommendations:

Major recommendations:

1) The amount of data shown in the manuscript is overwhelming. It would help the reader to focus on relevant data for the manuscript “take-home message” and include more details with all necessary controls while omitting some parts which seem less conclusive.

We would like to thank the reviewer for these comments concerning our manuscript. According to the reviewer's comment, we have carefully revised our manuscript, reducing the amount of data shown. We have excluded the data for microglia and astrocytes, focusing on the role of neuronal SphK1 (i.e., data for microglia in Supplementary Figure 1 and Supplementary Figure 5c, and data for microglia and astrocytes in Supplementary Figure 5f, Supplementary Figure 7, and Supplementary Figure 8 in the original manuscript). Further, as suggested by the reviewer, we have included more details regarding necessary controls (i.e., Supplementary Figure 1 and Supplementary Figure 13a in the revised manuscript). For example, in the Supplementary Figure 1, we added the data regarding the characterization of the isolated neurons, microglia, and astrocytes.

Original manuscript

Revised manuscript

Supplementary Figure 1

Figure 1e

Supplementary Figure 5c

Deleted

Supplementary Figure 5f

Supplementary Figure 5e

Supplementary Figure 7

Deleted

Supplementary Figure 8

Deleted

Additional Figure 1. Excluded data of microglia and astrocyte.

2) The authors indicate (page 32):

“The purity of the fractionated neurons and microglia were confirmed by neuronal (MAP2) and microglial (Iba1) markers..”.

This is an important control and should be shown as supplementary data for all cell populations (neurons, microglia and astrocytes).

We thank the reviewer for this pertinent comment. As suggested by the reviewer, we have carefully revised our manuscript. In our study, we isolated neurons and microglia from the cortex of mice using density gradient centrifugation. The isolated cells were confirmed to possess neuronal or microglial markers using immunofluorescence (Leon J et al., Sci Rep. 2016;6:22086; J Biol Methods. 2014;1(2):e11). The purity of the fractionated neurons and microglia was confirmed by the presence of neuronal (NeuN) and microglial (Iba1) markers. The neurons stained only with NeuN, and microglia stained only with Iba1, indicating that neurons and microglia were efficiently isolated by the method used (Supplementary Figure 1a). Further, the GLAST-1⁺CD11b⁻ cell population was sorted as astrocytes using Arial II (BD Sciences). The purity of the fractionated astrocytes was confirmed by determining the transcript levels of *GFAP* in the GLAST-1⁺CD11b⁻ cell population compared to that in other populations, using previously described methods (Neurobiol Aging. 2014 Jan;35(1):1-14.; Nat Commun. 2016 Apr 21;7:11295). The GLAST-1⁺CD11b⁻ cell population showed expression of *GFAP*, but the negative fraction (Rest) containing unstained cells, showed very low expression of *GFAP* (Supplementary Figure 1b). Moreover, we confirmed the purity of the fractionated astrocytes by immunolabeling for astrocyte markers (*GFAP*, *S100*, and *SOX2*), using previously described methods (J Vis Exp. 2013 Jan 19;(71); J Neuroinflammation. 2017 May 5). The astrocytes showed a positive signal when stained with antibodies against *GFAP*, *S100*, and *SOX2* (Supplementary Figure 1c). Based on these data, we conclude that the astrocytes were efficiently isolated and purified. To emphasize this point, the data regarding the characterization of the isolated neurons, microglia, and astrocytes have been presented in Supplementary Figure 1.

Supplementary Figure 1 in revised manuscript. Cortical neurons, microglia, and astrocytes are isolated from adult WT mice. **a**, Purity of the cortical neurons and microglia isolated from adult WT mice. Neuronal (up, NeuN, green; Scale bars, 200 μm) and microglial (down, Iba1, red; Scale bars, 50 μm) markers were detected using immunofluorescence. Nuclei were counter-stained by 4',6-diamidino-2-phenylindole (DAPI, blue). **b**, Transcript levels of *GFAP* in $\text{GLAST-1}^+\text{CD11b}^-$ cell population and the remaining population ($n = 4$ per group). **c**, Immunolabeling for astrocyte markers (*GFAP*, *S100*, and *SOX2*). Scale bar, 50 μm . **b**, Student's *t* test, $*P < 0.05$. All error bars indicate s.e.m.

3) *SphK1* immunofluorescence analysis (Fig 1e) shows an increase in neuronal *SphK1* signal in *APP/PS1/SphK1* tg and *SphK1* tg mice. Although a clear difference between WT and *APP/PS1* can be observed, an increase between WT and *APP/PS1/SphK1* tg and *SphK1* tg is more prominent in quantification analysis (Fig 1f) than reflected on the representative images. It would be good to provide western-blot analysis for *SphK1* levels in the brain, ideally in different cellular populations.

We would like to thank the reviewer for these comments concerning our manuscript. According to the reviewer's comment, we performed western blotting for *SphK1* in neurons, microglia and astrocytes derived from WT, *APP/PS1*, *APP/PS1/SphK1* tg and *SphK1* tg mice. The expression of *SphK1* was decreased in *APP/PS1* neurons, and increased in *APP/PS1/SphK1* tg and *SphK1* tg neurons compared with WT neurons, similar to the results for the quantification of neuronal *SphK1* immunofluorescence. In the revised manuscript, we have added this data in Figure 1g. The *SphK1* levels in microglia and astrocytes did not differ between WT and *APP/PS1* mice, while they were increased in *APP/PS1/SphK1* tg and *SphK1* tg compared with WT mice (Additional Figure 2). These data indicate that *SphK1* was decreased in *APP/PS1* neurons and increased in *APP/PS1/SphK1* tg and *SphK1* tg than WT mice.

Figure 1g in revised manuscript. g, Western blot analysis for SphK1 in neurons isolated from mouse brain (n = 4 per group). g, One-way analysis of variance, Tukey's post hoc test. * $P < 0.05$, ** $P < 0.01$. All error bars indicate s.e.m.

Additional Figure 2. Western blotting for SphK1 in microglia (a) and astrocytes (b) isolated from mouse brain (n = 4 per group). a-b, One-way analysis of variance, Tukey's post hoc test. * $P < 0.05$. All error bars indicate s.e.m.

4) I would suggest including Suppl Fig 2c as part of the Fig 2 to pinpoint the influence of SphK1 on the reduction of amyloid plaque burden and correlate this effect with microglial morphology. A β immunofluorescence analysis of amyloid plaques (for example using 6E10 antibody) should be included. High magnification images of microglia around amyloid plaques should also be presented.

According to the reviewer's comment, we have moved **Supplementary Figure 2c** to **Figure 2a** to point out the influence of SphK1 on the reduction of amyloid plaque load and correlate this effect with neuroinflammation. Also, as suggested by the reviewer, we performed additional immunofluorescence analysis of A β using 6E10 antibody. We confirmed that APP/PS1/SphK1 tg mice showed significantly lower A β levels than APP/PS1 mice in the 6E10 immunofluorescence analysis, similar to A β 40 and A β 42 level confirmed in the original manuscript (**Figure 2b**). In the revised manuscript, we have added this data in **Figure 2b**.

Figure 2b in revised manuscript. b, Representative immunofluorescence images and quantification of 6E10 (n = 6 per group; Scale bars, 80 μ m). b, Student's t test. ** P < 0.01, *** P < 0.001. All error bars indicate s.e.m.

In Figure 2, we show that overexpression of SphK1 improved the inflammatory response in the brain of AD mice. The regulation of neuroinflammation might be an important mechanism that attenuates AD pathology by increasing neuronal SphK1. Subsequently, we investigated the specific mechanism of neuroinflammation involved, and confirmed that neuronal SphK1 promoted the secretion of SPMs, especially 15-R-LxA4, which increases the number of phagocytic microglia (Figure 3-4). Thus, we presented the data on microglial A β phagocytosis

in Figure 5, and the results show increase of the phagocytic microglia in APP/PS1/SphK1 tg mice compared with APP/PS1 mice. According to the reviewer's comment, we have also replaced new high-magnification images of microglia surrounding amyloid plaques, which indicate microglial recruitment to the amyloid plaques and their phagocytotic ability, in **Figure 5a** of the revised manuscript. Based on these concepts, we consider these images to be more appropriate for Figure 5 than Figure 2. To clarify our results, we have added the new high magnification images of microglia surrounding amyloid plaques in **Figure 5a** of the revised manuscript.

Figure 5a in revised manuscript. a, Colocalization of microglia (Iba1, red) with Aβ (Aβ 42, green) and quantification. Scale bars, 10 μm; 3D reconstruction from confocal image stacks scale bars, 10 μm. (n = 7 per group). a, Student's t test. *P < 0.05. All error bars indicate s.e.m.

5) A decrease in levels of synaptophysin, MAP2, synapsin and PSD95 in APP/PS1 mice (Suppl Fig 2h-k) is surprisingly strong. This should be re-evaluated and discussed in the light of previous reports that show no neuronal loss in the APP/PS1 mouse line (Howlett et al., Brain Research, 2004). In addition, synaptophysin staining pattern in WT animals is unusual and completely different in Suppl Fig 2h and Suppl Fig 15f. Similarly, caspase 3 staining (Suppl Fig 4a) also displays an unusual staining pattern that rather resembles plaque-like structures in APP/PS1 mice. I would recommend evaluating the levels of synaptic proteins, MAP2 and caspase 3 by western-blotting. Of note, synaptophysin, MAP2, synapsin or PSD95 quantifications should not be indicated as burden.

We would like to thank the reviewer for these comments. However, we would also respectively point out that while neuronal loss in APP/PS1 mice was not mentioned in original study generating the line (Brain Res. 2004 Aug 13;1017(1-2):130-6.), another study by the author reported neuronal loss in the APP/PS1 mice (Histol Histopathol. 2008 Jan;23(1):67-76). Further, many studies have reported decreased levels of synaptophysin, MAP2, synapsin1

and PSD95 in APP/PS1 mice (Neural Regen Res. 2016 Oct;11(10):1617-1624; Nat Neurosci. 2013 Sep;16(9):1299-305). Moreover, similar results were also confirmed in our study. Based on the previous studies and our results, we conclude that neuronal loss may be observed in APP/PS1 mice.

According to the reviewer's comment, we have carefully reviewed all of our immunofluorescence images and data. As mentioned by the reviewer, the results in **Supplementary Figure 2h** and **Supplementary Figure 14f** (Supplementary Figure 15f in the original manuscript) may show different staining pattern in WT animals. However, this difference might be caused by the difference in age (i.e., Supplementary Figure 2h was done at 9-mo-old mice and Supplementary Figure 14f was done at 2-mo-old mice) and magnification (Supplementary Figure 2h; 10 μ m and Supplementary Figure 14f; 50 μ m). These results indicate that the levels of synaptophysin were reduced in 9-mo-old mice (**Supplementary Figure 2h**) rather than 2-mo-old mice (**Supplementary Figure 14f**). Similar results from previous studies show that the level of synaptophysin reduces with aging (Sci Rep. 2017 Sep 22;7(1):12196; Neural Regen Res. 2013 Oct 15;8(29):2725-33). Also, we represented high magnification image (scale bar, 10 μ M) from 9-mo-old mice (**Supplementary Figure 2h**) and low magnification image (scale bar, 50 μ M) from 2-mo-old mice (**Supplementary Figure 14f**) to show the apparent variation between the groups. Further, we wish to point out that many studies have reported images showing similar Synaptophysin staining to these (BMC Neurosci. 2009 Dec 4;10:144; Acta Neuropathol. 2013 Sep;126(3):329-52; Neural Plast. 2015;2015:374520). Due to the aforementioned reasons, we consider our results to be reliable.

As mentioned by the reviewer, the caspase 3 staining (Supplementary Figure 4a) appears unusual, resembling the plaques observed in APP/PS1 mice. However, we wish to point out that we have reviewed the literature and found that many studies have reported similar staining images for caspase3 in the brain of AD mice (Sci Rep. 2016 Dec 12;6:38452; Neurobiol Dis. 2016 Feb;86:29-40; Mol Psychiatry. 2015 Nov;20(11):1301-10; Brain Sci. 2017 Oct 14;7(10); Cell. 2017 Aug 10;170(4):649-663). Due to the aforementioned reasons, we consider our results to be reliable.

Further, as suggested by the reviewer, we also performed western blotting for synaptophysin, MAP2, synapsin 1, PSD95 and caspase3 in WT, APP/PS1, APP/PS1/SphK1 tg and SphK1 tg mice. We confirmed that the expression of synaptic markers was decreased in APP/PS1 compared with WT mice, and was restored in APP/PS1/SphK1 tg mice (**Additional Figure 3a-**

d). The level of caspase3 was increased in APP/PS1 compared with WT mice, while no significant difference was observed between APP/PS1 and APP/PS1/SphK1 tg mice (Additional Figure 3e). Similar results were observed upon immunofluorescence-based analysis of the synaptic markers and caspase3 in WT, APP/PS1, APP/PS1/SphK1 tg and SphK1 tg mice (Supplementary Figure 2h-k and Supplementary Figure 4a). Taken together, our results indicated that the levels of synaptic markers were restored in the APP/PS1/SphK1 tg mice, compared with APP/PS1 mice, and apoptosis was not associated with the role of increased SphK1 levels in attenuating AD pathology. If the reviewer considers this data to be relevant for publication, we will add the data as Supplementary Figure 2h-k and Supplementary Figure 4a.

Additional Figure 3. Western blotting and quantification of Synaptophysin (a), MAP2 (b), Synapsin1 (c), PSD95 (d), and Caspase3 (e) in the mouse brain (n = 4 per group). All data analysis was done at 9-mo-old mice. a-e, One-way ANOVA, Tukey's post hoc test. * $P < 0.05$, * $P < 0.01$. All error bars indicate s.e.m.

6) The authors should comment on visible differences in the response of microglia to A β treated WT neurons in Suppl Fig 6a and c. If such experimental variations are to be observed it would be hard to judge the protective effect of SphK1 overexpression.

According to the reviewer's comment, we performed additional experiments to confirm the visible differences in the response of microglia to A β treated WT and SphK1 tg neurons. Neurons derived from WT and SphK1 tg mice were stimulated with A β , and the CM of following neurons was treated to microglia. Morphology of activated microglia was observed in microglia treated with CM from A β -treated WT neurons compared with control neurons, and was reduced in microglia treated with CM from A β -treated SphK1 tg neurons (Supplementary Figure 6e). Similar results were observed in analysis of inflammatory cytokines of microglia treated with CM from A β -treated neurons (Supplementary Figure 6c). Based on these data, we conclude that the overexpression of neuronal SphK1 have the protective effect on microglia in the presence of A β . To clarify our results, we have added these data in Supplementary Figure 6e of the revised manuscript.

Supplementary Figure 6e in revised manuscript. e, Morphology of microglia treated with CM from Aβ-treated WT and SphK1 tg neurons. Up, imaris-based three-dimensional reconstruction of a representative microglia. Scale bars, 5 μm. Down, Imaris-based automated quantification of microglial morphology (n = 5-6 per group). e, One-way ANOVA, Tukey's post hoc test. *P < 0.05, **P < 0.01. All error bars indicate s.e.m.

7) Morphological characterization of microglia (Suppl Fig 7b) would benefit from additional analysis such as microglial number per plaque volume and microglial volume (or soma size). The authors stated on page 19 of the manuscript that microglial recruitment to amyloid plaques was analyzed in Fig. 5a. However, the figure itself refers to Aβ containing/Iba1-positive cells per plaque. From the presented analysis (Fig 5) it is hard to judge whether microglia really took up Aβ. As this result is of importance for the major conclusion on phagocytic activity, it is necessary to show high magnification images, best combined with 3D rendering analysis, to

strengthen this result. Similarly, immunofluorescent images of a phagocytosis assay presented in Fig 5b are very difficult to judge. Above suggested morphological characterization of microglia could help in correlating morphological changes with phagocytic capacity as amoeboid microglial morphology is indicative of phagocytic microglia.

According to the reviewer's comment, we performed additional analysis for morphological characterization of microglia surrounding A β plaques in APP/PS1 and APP/PS1/SphK1 tg mice. Morphology of activated microglia was observed in APP/PS1 mice, while phagocytic morphology (amoeboid microglial morphology) was observed in APP/PS1/SphK1 tg mice (Figure 5d). To clarify our results, we added these data in Figure 5d of the revised manuscript.

Figure 5d in revised manuscript. d, Morphology of microglia (Iba1, red) surrounding A β (ThioS, green) in cortex of APP/PS1 and APP/PS1/SphK1 tg mice. Up, high magnification (Scale bars, 10 μm) and imaris-based three-dimensional images (Scale bars, 5 μm) of microglia surrounding A β . Down, Imaris-based automated quantification of microglial morphology (n = 7-8 per group). Student's t test. * $P < 0.05$. All error bars indicate s.e.m.

According to the reviewer's comment, we have carefully reviewed all data on microglial phagocytosis, and performed new experiments to present apparent image of phagocytosis. We have replaced new high magnification and 3D images of microglia surrounding amyloid plaques in **Figure 5a** of the revised manuscript, and corrected the representative immunofluorescence images of a phagocytosis assay presented in **Figure 5b**. Further, as suggested by the reviewer, morphological characterization of microglia was supported by 3D rendered images, added to **Figure 5d** of the revised manuscript. These results indicate that the microglia of APP/PS1/SphK1 tg mice had more phagocytic morphology and increased phagocytotic ability than those of APP/PS1 mice. We thank the reviewer for helping us report our findings in a more coherent and precise manner.

Figure 5 in revised manuscript. Increased SphK1 improves microglial phagocytosis. a,

Colocalization of microglia (Iba1, red) with A β (A β 42, green) and quantification. Scale bars, 10 μ m; 3D reconstruction from confocal image stacks scale bars, 10 μ m (n = 7 per group). b, Left, representative photomicrograph of live slice section incubated with fluorescent beads (green). Scale bar, 10 μ m. White arrow point to phagocytotic microglia with fluorescent beads. Right, quantification of the number of microglial phagocytes normalized to the total number of microglia (n = 4-6 per group). c, Up, immunofluorescence images of thio S (A β plaques, green) encapsulated within Lamp1+ structures (phagolysosomes, blue) in microglia (Iba1, red) present in brains of APP/PS1 and APP/PS1/SphK1 tg mice. Low-magnification scale bars, 50 μ m; High-magnification scale bars, 10 μ m; 3D reconstruction from confocal image stacks scale bars, 10 μ m. Down, quantitation of microglial volume occupied by Lamp1+ phagolysosomes, percent of microglia containing A β -loaded phagolysosomes and A β encapsulated in phagolysosomes (n = 5 per group). d, Morphology of microglia (Iba1, red) surrounding A β (ThioS, green) in cortex of APP/PS1 and APP/PS1/SphK1 tg mice. Up, high magnification (Scale bars, 10 μ m) and imaris-based three-dimensional images (Scale bars, 5 μ m) of microglia surrounding A β . Down, Imaris-based automated quantification of microglial morphology (n = 7-8 per group). e, Morphometric analysis of A β plaques in APP/PS1 and APP/PS1/SphK1 tg mice (n = 5-6 per group). Brain sections were labeled with thio S and plaques were counted and assigned to three mutually exclusive size categories based on maximum diameter: small < 25 μ m; medium 25-50 μ m; or large > 50 μ m. All data analysis was done at 9-mo-old mice. a and c-e, Student's t test. b, One-way analysis of variance, Tukey's post hoc test. *P < 0.05, **P < 0.01, ***P < 0.001. All error bars indicate s.e.m.

8) *Suppl Fig 10b shows effects of SphK1 siRNA in neurons on secretion of various SPMs. However, the authors stated before (page 15) that only 15-R-LxA4 can be detected in neurons by LC-MS/MS:*

"Interestingly, only 15-RLxA4, one of COX2 acetylation-derived products, were found in neurons".

We would like to thank the reviewer for these comments concerning our manuscript. In this study, we examined whether neuronal SphK1 induces the secretion of the SPMs regulating microglial activation, using ELISA. The LxA4 and RvE1 levels were significantly lower in the CM derived from APP/PS1 neurons than that derived from WT neurons, and was restored in the CM derived from APP/PS1/SphK1 tg neurons, suggesting that neuronal SphK1 might be related with the secretion of SPMs (Figure 3a). However, LC-MS/MS has been used more widely than SPM ELISA kits, in previous studies, to detect and identify SPMs (Nature. 2007

Jun 14;447(7146):869-74; Blood. 2013 Jul 25;122(4):608-17), and is known to be more prefer method. According to these references, we performed LC-MS/MS based investigation of various SPMs, including 15-S-LxA4, 15-R-LxA4, 17-S-RvD1, 17-R-RvD1, 18-R-RvE1, and PD1. We confirmed that only 15-R-LxA4 was found in neurons, and that the neurons derived from APP/PS1 mice exhibited a marked decrease in the 15-R-LxA4 levels compared with those derived from WT mice. These were restored in the neurons derived from APP/PS1/SphK1 tg mice (Figure 3e and f). It is noteworthy that 15-S-LxA4, 17-S-RvD1, 17-R-RvD1, 18-R-RvE1 and PD1 were not detected in this LC-MS/MS analysis (Additional Figure 4). Since LC-MS/MS is based on the exact molecular formula rather than a fragmentation pattern, the SPM levels measured by LC-MS/MS were significantly lower, or undetectable, than the levels measured by ELISA. Due to this reason, we believe that RvE1 was not detected by LC-MS/MS even though ELISA showed that RvE1 levels were lower in APP/PS1 mice than in WT mice. Further, similar results, which showed reduction of LxA4 and RvE1 by ELISA, and only reduction of 15-R-LxA4 by LC-MS/MS, were observed in *SphK1* siRNA treated neurons (Supplementary Figure 8b and d). Based on previous studies and our results, we only stated the results for neuronal 15-R-LxA4 obtained by LC-MS/MS.

Additional Figure 4. Representative chromatograms of blank, internal standard (15-S-LxA4-d5), standard for individual SPMs and sample (neurons derived from WT mice). **a**, 17-R-RvD1 and 17-S-RvD1 were not detected in neurons. **b**, 15-R-LxA4 was identified, but not 15-S-LxA4. **c,d**, 18-R-RvE1 and PD1 were not detected in WT neurons.

9) *The authors analyzed A β phagocytosis in CamK2-cre^{+/-};SphK1^{flox/flox} mice after A β cortical injection (Suppl Fig 13). It is unclear what do the authors mean by Iba1-positive cells per plaque (Suppl Fig 13b) or area occupied by plaques (Suppl Fig 13d) as A β injection results in inducing amyloid plaque pathology only when injected in APP/PS1 background. Immunofluorescent images from the day of the injection (0) and analysis (7) should be shown including immunostaining for amyloid and microglia.*

We would like to thank the reviewer for these comments concerning our manuscript. We wish to point out that A β has previously been injected in various mouse models, including APP/PS1 background, to investigate amyloid plaque pathology (Sci Rep. 2015 Jun 29;5:11708; Science. 2016 May 6;352(6286):712-716; Front Aging Neurosci. 2016 Jan 11;7:245). Further, we consider it important to demonstrate the direct interaction of the specific decrease in neuronal SphK1 with microglial A β phagocytosis in order to reveal the role of neuronal SphK1 in AD environment. Accordingly, we injected A β into the cortex of CamK2-cre;SphK1^{flox/flox} mice, and examined whether the microglia affected by reduction of neuronal SphK1 phagocytosed the injected A β in CamK2-cre;SphK1^{flox/flox} mice. The results showed that although microglia surrounding the plaques did not exhibit any difference between control and CamK2-cre;SphK1^{flox/flox} mice, the injected A β was retained to a larger extent in the cortex of CamK2-cre;SphK1^{flox/flox} mice than that of control mice, suggesting that SphK1 deficiency in neurons interferes with amyloid uptake.

Further, as suggested by the reviewer, we performed an additional experiment to acquire immunofluorescence images of A β and microglia on the day of the injection (day 0) and analysis (day 7). Microglial recruitment to A β showed no significant differences on the day of injection (day 0) and analysis (day 7) between the groups (Additional Figure 5a-b). The injected A β in control mice was reduced after 7 days of injection compared with day 0, while the levels of injected A β in CamK2-cre;SphK1^{flox/flox} mice did not differ between day 0 and day 7 of the injection, indicating impaired microglial A β phagocytosis caused by knockout of neuronal SphK1 (Additional Figure 5c). However, we would also respectively point out that the immunofluorescence images of microglia surrounding A β alone cannot functionally judge phagocytotic ability. Therefore, in Supplementary Figure 12, we have investigated additional experiments including A β phagocytic aptitude, amount of remained A β and A β plaque morphometric analysis. The results show that CamK2-cre;SphK1^{flox/flox} mice exhibited a decrease of A β phagocytotic activity. Taken together, these data indicate that neuronal SphK1 deficiency reduced microglial phagocytosis.

Additional Figure 5. A β injection and analysis of CamK2-cre;SphK1^{flox/flox} mice in the day of the injection (0 day) and analysis (7 day). a, Immunofluorescent images for A β and microglia in the day of the injection (0 day) and analysis (7 day), Scale bars, 10 μ m. b, Quantitation of colocalization of microglia with A β in the day of the injection (0 day) and analysis (7 day) (n = 4 per group). c, Immunofluorescent images and quantitation of area occupied by A β plaques in brain of control and CamK2-cre;SphK1^{flox/flox} mice after A β cortical injection. (n = 4 per group) in the day of the injection (0 day) and analysis (7 day), Scale bars, 50 μ m. b-c, One-way analysis of variance, Tukey's post hoc test. *P < 0.05. All error bars indicate s.e.m.

10) Why do the numbers of phagocytes/Iba1-positive cells in control mice differ so much in Figures 5b and Suppl Fig 12e? Similar is also observed when comparing C14 signal in Suppl Figs 11a and 17c with Suppl Fig 14a.

According to the reviewer's comment, we have carefully reviewed all of our data. As mentioned by the reviewer, the results in Figure 5b and Supplementary Figure 11e (Supplementary Figure 12e in the original manuscript) show different numbers of phagocytes/Iba1-positive cells in control mice. This difference might be caused by the difference in age (i.e., Figure 5b was done at 9-mo-old mice and Supplementary Figure 11e was done at 2-mo-old mice). In our study, we confirmed that phagocytosis was lower, and the number of Iba1-positive cells was higher in 9-mo-old mice (Figure 5b) than in 2-mo-old mice (Supplementary Figure 11e). These results indicated greater microglial activation and lower microglial phagocytosis in 9-mo-old mice (Figure 5b) than in 2-mo-old mice (Supplementary

Figure 11e). These results are similar to those of previous studies reporting that microglia were activated, and their phagocytotic ability was reduced in the brains of aged mice (Curr Opin Pharmacol. 2016 Feb;26:96-101; Front Aging Neurosci. 2015 Apr 29;7:57). Thus, the numbers of phagocytes/Iba1-positive cells shown in Supplementary Figure 11e was higher than that shown in Figure 5b, and presented in different range in control mice.

Further, according to the reviewer's comment, we carefully reviewed our results, and found that the normal range of COX2 acetylation was about 6000-8000. Although, the range shown in Supplementary Figure 9a and 16c (Supplementary Figure 11a and 17c in the original manuscript) was within the normal range, the range shown in Supplementary Figure 13a (Supplementary Figure 14a in the original manuscript) was about 1500-2000, which was unusual. We speculated that this unusual range might be caused by the decreased of radioactivity of the C¹⁴ acetyl CoA in the previous experiment. Thus, we re-investigated COX2 acetylation in CamK2-cre;SphK1^{flox/flox} mice. We reconfirmed that SphK1-mediated COX2 acetylation was indeed decreased in neurons derived from CamK2-cre;SphK1^{flox/flox} mice, and was restored after 740YP treatment. Moreover, the C¹⁴ radioactivity range was within the normal range (Supplementary Figure 13a). To clarify our results, we have added these data in Supplementary Figure 13a of the revised manuscript. We thank the reviewer for helping us make our findings more coherent and precise.

Supplementary Figure 13a in revised manuscript. a, Acetylation assay of COX2 protein in neurons derived from control and CamK2-cre;SphK1^{flox/flox} mice with or without 740YP treatment (2 h after 10 μ M 740YP treatment). The [¹⁴C] aspirin treated neurons were used positive control. Scintillation counting of acetylation assay was performed using [¹⁴C] acetyl-CoA and COX2 protein in neurons (n = 6-8 per group). a, One-way ANOVA, Tukey's post hoc test. ***P* < 0.01. All error bars indicate s.e.m.

Minor recommendations:

1) *Consider including a more general reference for microglial dysfunction in AD (page 3). Additional reference is recommended for the evidence that NSAIDs may reduce risk of AD pathology (page 28).*

As suggested by the reviewer, we have cited Salter MW., Stevens B. 2017. Nat Med. as a more general reference for microglial dysfunction in AD in the Introduction section of the revised manuscript. Further, we have cited McGeer, P.L., McGeer, E., Rogers, J. & Sibley, J. 1990. Lancet. and Hoozemans, J.J. & O'Banion, M.K. 2005. Curr. Drug Targets CNS Neurol. Disord. to point out that NSAIDs may reduce the risk of AD pathology in the Discussion section of the revised manuscript.

2) *I would avoid using the M1 and M2 terminology throughout the manuscript as this classification has not proven to be useful.*

According to the reviewer's comment, we have carefully reconsidered our manuscript. We have reviewed the literature and found that many studies have used the phrases "activated microglia" and "phagocytic microglia" (Cell Stem Cell. 2010 Oct 8;7(4):483-95.; Front Cell Neurosci. 2013 Jan 30;7:6.; Cell. 2017 Jun 15;169(7):1276-1290.e17.). Accordingly, we have replaced the terms M1 and M2 with "activated microglia" and "phagocytic microglia", respectively, in the revised manuscript.

3) *It would be helpful to indicate SphK mRNA (fold change) and SphK activity on the figure axes such as Fig. 1a to distinguish better between the two experimental paradigms.*

According to the reviewer's comment, we have made the necessary changes in the revised manuscript.

4) *The authors state on page 6:*

"However, this activity was slightly decreased in cells derived from SphK2^{-/-} mice".

This decrease is obvious and significant and slightly may not be the best description.

According to the reviewer's comment, we have corrected this sentence in the revised manuscript. We have revised this sentence to: "SphK activity was decreased in *SphK1*^{-/-} and *SphK2*^{-/-} mice-derived neurons and increased in SphK1 tg mice-derived neurons compared with neurons derived from WT mice. Particularly, SphK activity in *SphK1*^{-/-} mice-derived neurons was lower than *SphK2*^{-/-}, indicating SphK1 was more effective in SphK activity. "

5) Introduction of the SphK1 tg and APP/PS/SphK1 tg mice (page 6) should come before as it is part of the Fig 1a. Consider introducing SphK1 phenotypes in all lines first and then showing the lipid analysis data. This would help the reader to keep the track.

As suggested by reviewer, we have carefully reviewed our manuscripts. We agree with the reviewer's comment that the SphK1 tg and APP/PS1/SphK1 tg mice should be introduced earlier, as these models are part of the Fig 1a. However, we wish to point out that we confirmed decreased neuronal SphK1 in APP/PS1 in comparison with WT mice, and want to focus on this difference between the WT and APP/PS1 mice. Thus, we described the difference between WT and APP/PS1 mice prior to introducing the SphK1 tg mice, and then introduced the breeding of SphK1 overexpressing tg mice with APP/PS1 animals to investigate whether increased neuronal SphK1 attenuates AD pathology. Therefore, we consider that original order of the results to be appropriate for focusing on the decreased levels of SphK1 in APP/PS1 neurons, and hope the reviewer understands our preference. Alternatively, we have merged the first two sections in the results of revised manuscript.

6) Consider modifying the following sentence (page 9) as IL-10 behavior is different compared to the rest of the anti inflammatory markers and is indeed decreased in APP/PS1/SphK1 mice compared to APP/PS1 mice.

"In addition, APP/PS1/SphK1 tg mice showed a decrease of pro-inflammatory M1 markers and immunoregulatory cytokines, and an increase of anti-inflammatory M2 markers compared with APP/PS1 mice".

The same applies to the description of Suppl Fig 7a on page 12.

According to the reviewer's comment, we have carefully reviewed our manuscript. However, we wish to point out that a recent study reported that IL10 was increased in APP/PS1 mice, and IL10 deficiency promotes Alzheimer's A β clearance in APP/PS1 mice (Neuron. 2015 Feb 4;85(3):534-48). We obtained the similar results in our study, with higher levels of IL10 in

APP/PS1 than WT mice, which were restored in APP/PS1/SphK1 tg mice. Thus, we described IL10 as an immunoregulatory cytokines rather than an anti-inflammatory cytokine to avoid confusion with other anti-inflammatory cytokines. Furthermore, we have reviewed the literature, and found that many studies had spelled out the pro-inflammatory mediators and anti-inflammatory mediators for a general readership (Nature. 2013 Jan 31;493(7434):674-8; Nat Med. 2014 Oct;20(10):1157-64; Nature. 2013 Feb 7;494(7435):90-4). Accordingly, we have spelled out the pro-inflammatory mediators and anti-inflammatory mediators in the revised manuscript.

7) SphK1 levels were analyzed in both neurons (Suppl Fig 5b) and microglia (Suppl Fig 5c), but microglial data are not described in the text (page 10). Suppl Fig 5f is showing SphK1 mRNA levels in neurons, but not in astrocytes or microglia while the activity has been analyzed in all cell types. The authors should be more consistent through the manuscript. For the reader it is sometimes hard to follow why certain cellular populations have been included and others not (additional example: Suppl Fig 1a).

According to the reviewer's comment, we have carefully reconsidered our manuscript. In our study, we isolated neurons, microglia and astrocytes, which are known to be important cell types related to AD pathogenesis. *Sphk1* mRNA expression and SphK activity were significantly decreased in APP/PS1 neurons compared with WT neurons, although there were no significant difference in microglia and astrocytes (Figure 1c). These results indicated that SphK1 was decreased only in APP/PS1 neurons, not in microglia and astrocyte. Accordingly, we excluded the data for microglia and astrocyte from Figure 1c to focus on elucidating the role of neuronal SphK1 for a general readership (i.e., data for microglia in Supplementary Figure 1 and Supplementary Figure 5c, data for microglia and astrocyte in Supplementary Figure 5f, Supplementary Figure 7 and Supplementary Figure 8 in the original manuscript).

8) Suppl Fig 5i is showing SphK1 mRNA levels in A β treated neurons and not as indicated in the text in AD brains (page 11).

According to the reviewer's comment, we have corrected this in the revised manuscript.

9) The following sentence (page 13) is not informative:

“AD-like pathology was also improved in 12-mo-old APP/PS1/SphK1 tg mice (data not shown), suggesting that the increase of SphK1 protects disease progression even at older ages”.

According to the reviewer’s comment, we have deleted this sentence in the revised manuscript.

10) *I would highly recommend streamlining and shortening the discussion.*

According to the reviewer’s comment, we have carefully reviewed our manuscript, and tried to shorten the length of discussion in the revised manuscript.

18) *It has been stated on page 27:*

“It is widely accepted that microglia degenerate during AD progression and shift to an M1 activation state, with loss of phagocytic function”.

Please, refer to minor recommendations/comment 2. In addition, microglial degeneration in AD has not been proven yet. This point should be discussed in the light of recent publications demonstrating that phagocytic capacity of AD microglia can be restored which should not be the case if overt degeneration of microglia would occur.

According to reviewer’s comment, we have carefully reconsidered our manuscript and agree with the reviewer’s comment that microglial degeneration in AD has not been proven yet. As suggested by the reviewer, we have reviewed the recent literature (Nat Med. 2017 Sep 8;23(9):1018-1027; Cell. 2017 Aug 10;170(4):649-663; Cell. 2017 Jun 15;169(7):1276-1290.e17) and discussed the phagocytic capacity of microglia in AD, relation to our study. We thank the reviewer for helping us improve the clarity of our study.

19) *Additional reference could be added to statement on page 27:*

“Recently, SPMs also have been shown to stimulate microglial phagocytosis and down-regulate neuroinflammation”.

Manuscript by Zhu et al., Mol Neurobiol, 2016 is addressing the role of SPMs in inflammation resolution pathway in AD. However, although they also provide evidence for important regulatory of SPMs is microglial phagocytosis, this beneficial effect was not exerted by LXA4 SPM.

As suggested by the reviewer, we have cited Medeiros R. et al., Am J Pathol. 2013 in the

Discussion section to point out that LxA4 stimulates microglial phagocytosis and down-regulates neuroinflammation.

On the reviewer's all comments concerning our manuscript, we would like to thank the reviewer once again.

Reviewer #3 (Remarks to the Author):

Ju Youn Lee and co-authors present a comprehensive set of data supporting a novel and disease relevant role for neuronal SphK1 in the regulation of microglial phagocytosis by modulating COX2-mediated “specialized proresolving mediators” (SPMs) secretion. The authors show that the underlying mechanism involves a SphK1 acetyltransferase activity towards COX2 and identified Ser565 as the targeted residue. Interestingly, the presented data show that neuronal SphK1 is reduced in Alzheimer’s disease (AD) patient brains/cells and this reduction lowers acetylated-COX2 levels and leads to decreased SPMs secretion. In contrast, SphK1 activation by 740YP treatment induces SPMs secretion. Therefore, the authors conclude that neuronal SphK1 acetyltransferase activity may contribute to AD pathology by impairing microglial phagocytosis and resolution of inflammation, and propose that activation of neuronal SphK1 acetyltransferase on COX2 as a potential novel therapeutic strategy in AD.

The study is highly relevant for the AD field and as discussed, may open novel therapeutic avenues. Furthermore, the elucidated mechanisms may have a broader significance in other CNS disorders.

Major comments:

1. “APP/PS1/SphK1 tg mice showed a significant reduction of microglia and astrocytes compared to APP/PS1 mice (Fig. 2a, b).” However, the authors also show that “microglial recruitment was increased in APP/PS1/SphK1 tg mice compared APP/PS1 mice (Fig. 5a).” and “More phagocytic microglia were found in APP/PS1/SphK1 tg mice compared APP/PS1 mice (Fig. 5b).”

How do the authors explain the differential effects on SphK1 (tg mice) on the size of the microglia population and phenotype.

We would like to thank the reviewer for these comments concerning our manuscript. A recent study has reported that microglia are dynamic cells that respond to a nearly endless variety of environmental cues, and a hallmark of microglial responsivity is the cells’ ability to alter their own morphology. Further, microglia have complex roles that are both beneficial and detrimental to disease pathogenesis, including engulfing or degrading toxic proteins and promoting neurotoxicity through excessive release of inflammatory cytokine. In neurodegenerative diseases, including Alzheimer’s disease, it is known that the number of

activated microglia, causing neuroinflammation, is increased, while that of phagocytic microglia, removing debris including A β , is reduced (Nat Med. 2017 Sep 8;23(9):1018-1027). In our study, the microglia, presented in **Figure 2**, indicated activated microglia, and their number was increased in APP/PS1 mice compare to WT mice, and was restored in APP/PS1/SphK1 tg mice. On the other hand, the microglia observed in APP/PS1/SphK1 tg, presented in **Figure 5**, indicated phagocytic microglia, and were higher in number than APP/PS1 mice. Taken together, all results indicate that increased neuronal SphK1 reduced the number of activated microglia, and increased the number of phagocytic microglia.

2. Neuronal SphK1 acetyltransferase activity depends on the presence of sphingosine and the authors suggest that acetylation may occur through a sphingosine intermediate.

the authors should evaluate how sphingosine changes the kinetic (K_m , k_{cat}) parameters of SphK1. This is relevant due to the potential therapeutic value of this activity.

According to the reviewer's comment, we have performed additional experiments for kinetic and functional analysis of acetyl-CoA binding to SphK1 in the presence of 0, 10 and 100 μ M sphingosine. Incorporation of the acetyl group into SphK1 was not observed for 0 μ M of sphingosine, and increased with increasing concentration of sphingosine, yielding the 60.2 μ M and 6.4 μ M K_M values at 10 μ M and 100 μ M sphingosine, respectively. However, K_{cat} did not differ with changing concentration of sphingosine (**Figure 4c**). These results indicated that both sphingosine and acetyl-CoA were needed for SphK1 mediated COX2 acetylation, suggesting that sphingosine may participate in this catalytic reaction.

Figure 4c. Acetyl-CoA binding activity of SphK1 was analyzed by filter binding assay in the presence of 0, 10 and 100 μM sphingosine. The binding velocity (V_{binding}) of [^3H] acetyl-CoA to SphK1 was plotted to the acetyl-CoA concentration and the nonlinear regression analysis of the saturated plot yielded the kinetic parameters such as K_{cat} (catalytic constant) and K_M (Michaelis-Menten constant) for acetyl-CoA and SphK1 binding activity in the presence of 0, 10 and 100 μM sphingosine ($n = 3$ per group).

Minor comments:

I suggest to merge and shorten the first two sections in the results and to improve clarity figure legends should be revised.

According to the reviewer's comment, we have merged and shortened the first two sections in the revised manuscript. Further, we have improved the clarity of all the figure legends as well as Figure1.

On the reviewer's all comments concerning our manuscript, we would like to thank the reviewer once again.

Reviewers' comments:

Reviewer #2 (Remarks to the Author):

The authors have partially addressed my criticism and improved the manuscript. Quality control (Suppl Fig 1) is convincingly shown for astrocytes, but not for neurons and microglia (morphology of neurons and microglia is not convincing; was the analysis, such as SphK mRNA, done on acutely isolated or cultured cells?). I agree with the authors that CamK2-Cre;SphK1flox/flox mice display decreased phagocytic activity towards injected A β (Suppl Fig 12), but what I tried to point out is that it is inappropriate to call the injected A β material plaque.

Consider modifying your altered sentence on page 3 of the manuscript and omit "decrease of activated microglia" and instead just describe your findings: increase of phagocytic microglia, downregulation of pro-inflammatory cytokines etc...Similarly on page 19 I would omit "Morphology of activated microglia was observed in APP/PS1 mice" (in the literature often activated microglia are indeed characterized by amoeboid morphology, thus amoeboid morphology is not indicative for decreased activation). It is more precise to state that you observed increase in amoeboid microglia morphology in APP/PS1/SphK1 tg mice compared to APP/PS1.

Consider showing only western blot data in Suppl Figs 2h-k and 4a. I would suggest excluding Suppl Fig 6. from the manuscript. I think that the authors provided enough convincing in vivo evidence that neuronal SphK1 deficiency influences microglia function and that rather artificial and complex cell culture experiments (with poorly described experimental details and controls) do not add any additional information.

Reviewer #3 (Remarks to the Author):

The authors have made an important revision that has undoubtedly improved the clarity of the manuscript. In addition, they have satisfactorily addressed the points raised by this reviewer.

Reviewer #2 (Remarks to the Author):

The authors have partially addressed my criticism and improved the manuscript. Quality control (Suppl Fig 1) is convincingly shown for astrocytes, but not for neurons and microglia (morphology of neurons and microglia is not convincing; was the analysis, such as SphK mRNA, done on acutely isolated or cultured cells?). I agree with the authors that CamK2-Cre;SphK1flox/flox mice display decreased phagocytic activity towards injected A β (Suppl Fig 12), but what I tried to point out is that it is inappropriate to call the injected A β material plaque. Consider modifying your altered sentence on page 3 of the manuscript and omit “decrease of activated microglia” and instead just describe your findings: increase of phagocytic microglia, downregulation of pro-inflammatory cytokines etc...Similarly on page 19 I would omit “Morphology of activated microglia was observed in APP/PS1 mice” (in the literature often activated microglia are indeed characterized by amoeboid morphology, thus amoeboid morphology is not indicative for decreased activation). It is more precise to state that you observed increase in amoeboid microglia morphology in APP/PS1/SphK1 tg mice compared to APP/PS1.

Consider showing only western blot data in Suppl Figs 2h-k and 4a. I would suggest excluding Suppl Fig 6. from the manuscript. I think that the authors provided enough convincing in vivo evidence that neuronal SphK1 deficiency influences microglia function and that rather artificial and complex cell culture experiments (with poorly described experimental details and controls) do not add any additional information.

Quality control (Suppl Fig 1) is convincingly shown for astrocytes, but not for neurons and microglia (morphology of neurons and microglia is not convincing; was the analysis, such as SphK mRNA, done on acutely isolated or cultured cells?).

We would like to thank the reviewer for these comments concerning our manuscript. In our study, we isolated neurons and microglia from the cortex of mice using density gradient centrifugation. The isolated neurons and microglia were acutely confirmed to possess neuronal or microglial markers using immunofluorescence and analyzed for *SphK* mRNA levels, SphK activity, and the acetylation assay. Based on this reasons, the image of isolated neurons and microglia might appears unusual compared with cultured neurons and microglia. However, the isolated neurons stained only with NeuN, and isolated microglia stained only with Iba1, indicating that neurons and microglia were efficiently isolated by the method used (Supplementary Figure 1a). Further, we wish to point out that we have reviewed the literature using identical method that we used to isolate cells and found that many studies have reported similar staining images for isolated neuron and microglia in the adult brain (Sci Rep. 2016 Feb 25;6:22086; Sci Rep. 2015 Nov 18;5:16763; BMC Neurosci. 2013 Oct 4;14:112; Front Cell Neurosci. 2014 Jun 2;8:152). Due to the aforementioned reasons, we consider our results to be reliable. Also, we have added detailed description in Cell isolation of the methods section of the revised manuscript.

I agree with the authors that CamK2-Cre;SphK1^{flox/flox} mice display decreased phagocytic activity towards injected A β (Suppl Fig 12), but what I tried to point out is that it is inappropriate to call the injected A β material plaque.

According to the reviewer's comment, we have carefully reviewed our manuscript. We agree with reviewer's comment that it is inappropriate to call injected A β in CamK2-Cre;SphK1^{flox/flox} mice as the A β plaque, because A β injection results in inducing amyloid plaque pathology only when injected in APP/PS1 background. Accordingly, we have replaced "plaques" with "injected A β " or "A β " in the revised manuscript. We thank the reviewer for helping us improve the clarity of our study.

Consider modifying your altered sentence on page 3 of the manuscript and omit "decrease of activated microglia" and instead just describe your findings: increase of phagocytic microglia, downregulation of pro-inflammatory cytokines etc...Similarly on page 19 I would omit "Morphology of activated microglia was observed in APP/PS1 mice" (in the literature often

activated microglia are indeed characterized by amoeboid morphology, thus amoeboid morphology is not indicative for decreased activation). It is more precise to state that you observed increase in amoeboid microglia morphology in APP/PS1/SphK1 tg mice compared to APP/PS1.

According to the reviewer's comment, we have corrected these sentences in the revised manuscript. We have revised the sentence in page 3 of the manuscript to: "This results in restoration of microglia function, such as increase of phagocytic microglia, downregulation of pro-inflammatory cytokines, and active clearance of apoptotic cells and debris." Also, we have revised the sentence in page 19 of the manuscript to: "the phagocytic morphology (amoeboid microglial morphology) was more observed in APP/PS1/SphK1 tg mice than APP/PS1 mice."

Consider showing only western blot data in Suppl Figs 2h-k and 4a.

According to the referee's comment, we have replaced the immunofluorescence data with the western blot data in **Supplementary Figure 2h-k and 4a** of the revised manuscript.

I would suggest excluding Suppl Fig 6. from the manuscript. I think that the authors provided enough convincing in vivo evidence that neuronal SphK1 deficiency influences microglia function and that rather artificial and complex cell culture experiments (with poorly described experimental details and controls) do not add any additional information.

According to the reviewer's comment, we have carefully reviewed our manuscript. We partially agree with reviewer's comment that we provided enough convincing in vivo evidence that neuronal SphK1 deficiency influences microglia function. However we wish to point out that the sequence of microglial activation appears to be neurons to microglia directly, not through astrocytes using in vitro experiments in Supplementary Figures 6. Moreover, the editor also commented that they are fine with keeping Supplementary Figures 6 in the manuscript though we may elect to leave it out. Due to the aforementioned reasons, we consider to keep Supplementary Figures 6 to be appropriate and hope the reviewer understands this situation.

On the reviewer's all comments concerning our manuscript, we would like to thank the reviewer once again.

Reviewer #3 (Remarks to the Author):

The authors have made an important revision that has undoubtedly improved the clarity of the manuscript. In addition, they have satisfactorily addressed the points raised by this reviewer.

We thank the reviewer for these positive comments concerning our manuscript.